# In vivo self-renewal and expansion of quiescent stem cells from a non-human primate

Jengmin Kang [1,2,8,11], Abhijnya Kanugovi[1,2,11], M. Pilar J. Stella [3,11], Zofija Frimand [3], Jean Farup[1,2,3,9], Andoni Urtasun[1,2], Shixuan Liu [4,5], Anne-Sofie Clausen [3], Heather Ishak[1,2], Summer Bui[1,2], Soochi Kim[1,2,10], Camille Ezran [4], Olga Botvinnik[6], Ermelinda Porpiglia [3], Mark A. Krasnow [4], Antoine de Morree [1,2,3,9] ✉ & Thomas A. Rando [1,2,7,8] ✉

The development of non-human primate models is essential for the fields of developmental and regenerative biology because those models will more closely approximate human biology than do murine models. Based on single cell RNAseq and fluorescence-activated cell sorting, we report the identification and functional characterization of two quiescent stem cell populations (skeletal muscle stem cells (MuSCs) and mesenchymal stem cells termed fibro-adipogenic progenitors (FAPs)) in the non-human primate *Microcebus murinus* (the gray mouse lemur). We demonstrate in vivo proliferation, differentiation, and self-renewal of both MuSCs and FAPs. By combining cell phenotyping with cross-species molecular profiling and pharmacological interventions, we show that mouse lemur MuSCs and FAPs are more similar to human than to mouse counterparts. We identify unexpected gene targets involved in regulating primate MuSC proliferation and primate FAP adipogenic differentiation. Moreover, we find that the cellular composition of mouse lemur muscle better models human muscle than does macaque (*Macaca fascicularis*) muscle. Finally, we note that our approach presents as a generalizable pipeline for the identification, isolation, and characterization of stem cell populations in new animal models.

Adult stem cells are key for tissue maintenance and regeneration. Many tissues harbor populations of adult stem cells in a quiescent state, defined by a prolonged reversible withdrawal from the cell cycle, in reserve for the repair of tissue injuries. Much of our understanding of quiescent stem cells comes from studies in mice[1]. However, although mice are invaluable as a model for the study of conserved stem cell function, there is a pressing need for new animal models that more closely reflect human physiology[2,3]. This is certainly the case for

[1]Department of Neurology and Neurological Sciences, Stanford University School of Medicine, Stanford, CA, USA. [2]Paul F. Glenn Laboratories for the Biology of Aging, Stanford University School of Medicine, Stanford, CA, USA. [3]Department of Biomedicine, Aarhus University, Aarhus, Denmark. [4]Department of Biochemistry and Howard Hughes Medical Institute, Stanford University School of Medicine, Stanford, CA, USA. [5]Department of Chemical and Systems Biology, Stanford University School of Medicine, Stanford, CA, USA. [6]Chan Zuckerberg Biohub, San Francisco, CA, USA. [7]Neurology Service, Veterans Affairs Palo Alto Health Care System, Palo Alto, CA, USA. [8]Present address: Broad Stem Cell Research Center, University of California Los Angeles, Los Angeles, CA, USA. [9]Present address: Department of Biomedicine, Aarhus University, Aarhus, Denmark. [10]Present address: Department of Biotechnology and Bioinformatics, Korea University, Sejong, Republic of Korea. [11]These authors contributed equally: Jengmin Kang, Abhijnya Kanugovi, M. Pilar J. Stella. ✉e-mail: demorree@biomed.au.dk; trando@mednet.ucla.edu

stem cells, for which there are considerable differences in function across species even within the same tissue[4].

Non-human primates are of great interest as animal models due to the similarity of their genetics and physiology to humans. Adult stem cells have rarely been studied in non-human primates, at least in part due to a lack of conserved cell type markers and cross-reacting antibodies that recognize those markers. The CD34[+] cell fractions, characteristic of hematopoietic stem cells (HSCs) in mice, from the bone marrow of juvenile rhesus macaques show long-term engraftment potential in irradiated hosts[5,6]. Nevertheless, markers for the isolation of long-term quiescent hematopoietic stem cells remain elusive. In addition to the challenges with identifying stem cell populations come difficulties with studying non-human primates. Macaques are difficult and expensive to work with due to their large size and long life span.

Mouse lemurs are among the smallest (<90 g), fastest reproducing, and most abundant primates[7]. Their small size and similar physiology to humans make them an attractive candidate for the study of primate biology and pathophysiology. Here, we use single-cell analyses to identify and characterize quiescent stem cell populations in the gray mouse lemur *M. murinus*, and we demonstrate the applicability of mouse lemurs to uncover therapeutic targets for human diseases.

## Results
### Myogenic cells in dissociated mouse lemur muscle tissue
To identify stem cell populations in mouse lemur skeletal muscle, we dissected limb skeletal muscles, obtained postmortem from two aged mouse lemurs, and digested them into mononuclear cell suspensions. A portion of these cell suspensions was set aside for RNA sequencing, while the remainder was enriched for myoblast-like cells by pre-plating with the goal of performing a myogenic differentiation assay. After four weeks of cell expansion in growth medium, we were able to differentiate mouse lemur myoblast-like cells into contracting myotubes (Fig. 1a), demonstrating the presence of myogenic cells in the skeletal muscle cell suspension.

### Single-cell RNA sequencing profiles uncover putative stem cell populations
The cells retained for sequencing were subjected to droplet sequencing using 10X genomics technology. Sequence reads were aligned to the MicMur3lemur genome[8]. Cells with a minimum of 1000 reads were used for the analysis. In total, we analyzed 3122 single cells from one individual and 9409 single cells from the other individual. To identify cell populations, we performed a dimension reduction analysis of the 10X data and identified 22 distinct cell clusters (Fig. 1b, c). One cluster of cells was characterized by the expression of the myogenic transcription factor *MYF5* (Fig. 1d), suggesting that this cluster represented the mouse lemur MuSC population. However, canonical mouse MuSC markers, such as *PAX7*, *PAX3*, *CALCR*, *VCAM1*, and *ITGA7*, were not detectable. Commonly used antibodies (VCAM1, SCA1, CD45, CD31) for the purification of mouse MuSCs were not effective for labeling lemur MuSCs as determined by flow cytometry (Supplementary Fig. 1a, b).

Another cluster of cells was characterized by the expression of genes encoding matrix proteins and matrix remodelers, suggesting a fibrogenic cell type. A small subset of cells in this cluster expressed the canonical mesenchymal progenitor marker *PDGFRA* (Fig. 1d). As with the putative MuSCs, commonly used antibodies (SCA1) for the purification of mouse FAPs did not show any staining by flow cytometry (Supplementary Fig. 1a, b).

To support our tentative conclusion as to the identity of these two cell clusters, we identified the top 15 cell surface markers that defined each cluster. Cells in the *MYF5* cluster were found to express *NCAM1* (CD56) (Fig. 1e), which is a known marker of human MuSCs[9,10]. Furthermore, the expression of the gene *THY1* (CD90), which is a known marker of mouse and human mesenchymal progenitors[11,12], was detected in the putative mesenchymal progenitor cluster (Fig. 1e).

To further confirm the identity of these populations, we analyzed the transcriptomes of single cells purified by fluorescence-activated cell sorting (FACS) from a third aged mouse lemur and sequenced by SmartSeq2 (SS2) technology, which yields fewer cells but at a higher read depth[13]. We obtained 158 single-cell transcriptomes with a minimum of 5000 reads, with a comparable distribution of cell clusters to the 10X data (Supplementary Fig. 1c, d). We also identified a small cluster of *MYF5*-high cells in which *PAX7* was detected at low levels and in which *NCAM1* is among the top enriched cell surface markers (Supplementary Fig. 1c). Similarly, we identified a *PDGFRA*-enriched cell cluster in which *THY1* is among the top enriched cell surface markers (Supplementary Fig. 1d).

### Immunophenotyping of mouse lemur muscle mononuclear cell suspensions
To test whether NCAM1 and THY1 could be used as markers for the purification of MuSCs and mesenchymal progenitors, respectively from the mouse lemur, we dissected and digested limb skeletal muscles obtained postmortem from four aged mouse lemurs. Utilizing antibodies against human NCAM1 and THY1, we were able to identify and purify two distinct cell populations by FACS (Fig. 1f, Supplementary Fig. 1e). In contrast, staining with antibodies against human FGFR4, the top lemur MuSC marker, did not allow the identification of a distinct cell population by flow cytometry analysis (Supplementary Fig. 1e). The freshly isolated NCAM1[+]/THY1[-] cells stained positive with a monoclonal antibody against a conserved epitope in the PAX7 protein (~85% PAX7[+ve]) (Supplementary Fig. 1f, Supplementary Data 1). This percentage is similar to what we observed for purified mice and human MuSCs (approximately 90% and 85% PAX7[+ve], respectively). Furthermore, we observed PAX7[+] cells adjacent to myofibers in mouse lemur muscle cryosections (Fig. 1g). Similarly, the freshly isolated THY1[+]/NCAM1[-] cells stained positive with a monoclonal antibody against a conserved epitope in the PDGFRα protein (~80% PDGFRα[+ve]) (Supplementary Fig. 1g). This percentage is similar to what we observed for purified mouse and human FAPs (approximately 95% and 90% PDGFRα[+ve], respectively). Furthermore, we identified PDGFRα[+ve] cells interstitially in mouse lemur muscle cryosections (Fig. 1h). When we purified these cell populations by FACS and analyzed the single cells by SS2, we obtained 285 transcriptomes of the NCAM1[+]/THY1[-] cells, more than 80% of which had detectable levels of RNA of the MuSC markers *PAX7* and *MYF5*, a fraction that is similar to what we find in SS2 experiments with human and mouse MuSCs and likely reflecting that a substantial number of MuSCs express low numbers of PAX7 transcript below the detection limit[14–17]. We also obtained 255 transcriptomes of the THY1[+]/NCAM1[-] cells, ~75% of which had detectable levels of RNA of the FAP marker *PDGFRA* (Supplementary Data 2), demonstrating that the two molecularly defined cell types can be prospectively isolated on the basis of the two cell surface proteins NCAM1 and THY1.

To confirm that the NCAM1[+]/THY1[-] cells are highly enriched for bona fide myogenic cells, we seeded NCAM1[+]/THY1[-] single cells isolated from three mouse lemurs into a growth medium for a myoclone assay. We were able to clonally expand ~25% of single cells, a fraction similar to what we previously reported for geriatric mouse MuSCs[18]. All of these clones were able to differentiate into multinucleated myotubes that stain positive for the mature myofiber protein MYH2 (Fig. 1i, j). Moreover, when we seeded and expanded 20,000 NCAM1[+]/THY1[-] cells, we obtained progenitors that retained NCAM1 expression and readily differentiated into contracting myotubes that stain positive for MYH2 protein (Supplementary Fig. 1h–j, Supplementary Movie 1). We conclude that NCAM1 expression marks MuSCs in mouse lemur skeletal muscle.

To test whether the THY1[+]/NCAM1[-] cells are highly enriched for bona fide mesenchymal progenitors, we seeded THY1[+]/NCAM1[-] single cells into growth medium for a clonal assay. We were able to

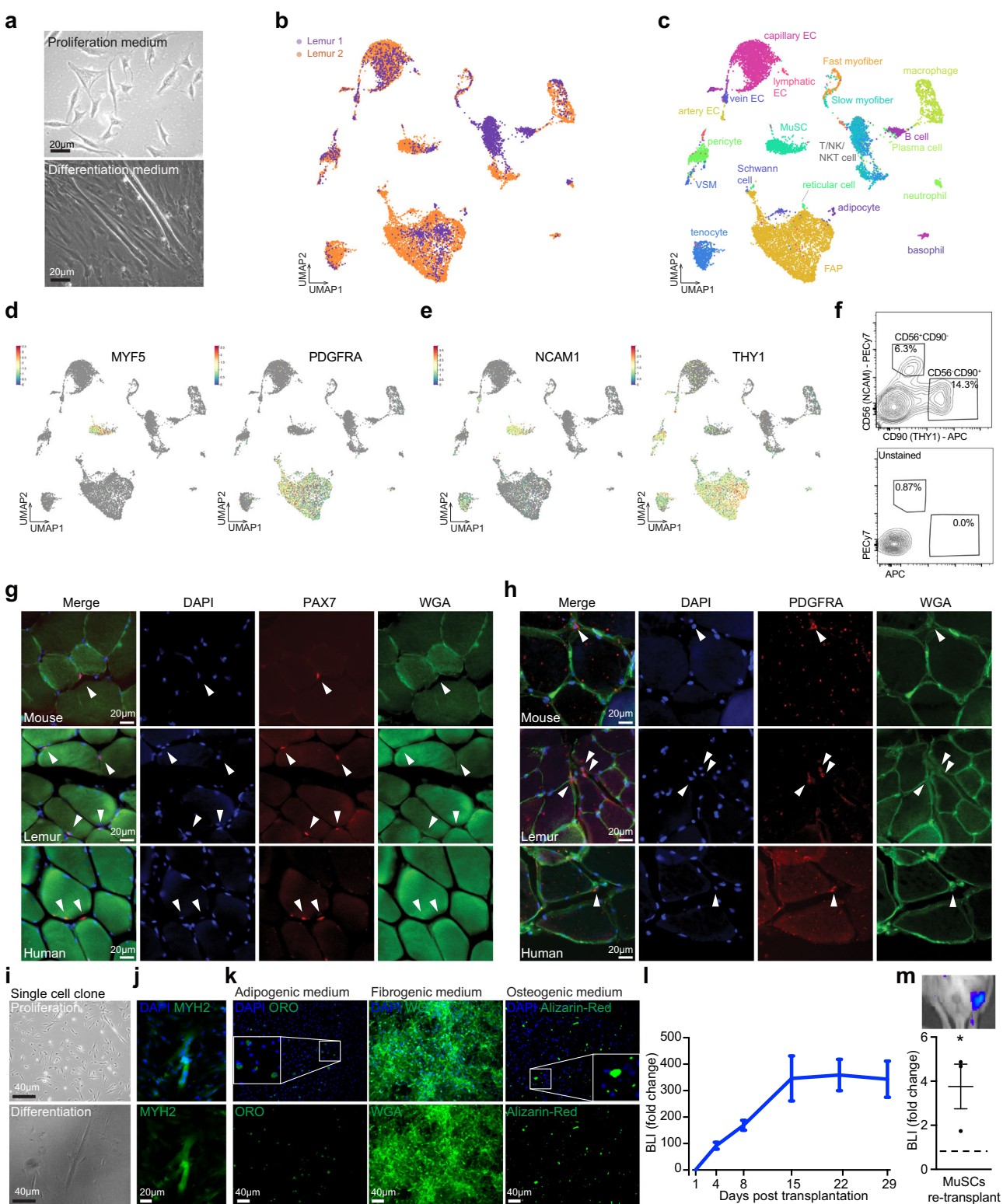

clonally expand ~20% of single cells. We next seeded 20,000 THY1+/NCAM1− cells and grew them in adipogenic, fibrogenic, or osteogenic medium[19,20]. Consistent with the multipotent nature of FAPs[19,20], we observed the formation of adipocyte-, fibroblast-, and osteocyte-like cells but not MyoD-positive myogenic cells (Fig. 1k, Supplementary Fig. 1k, l). We conclude that THY1+/NCAM1− cells in mouse lemur skeletal muscle have adipogenic, fibrogenic, and osteogenic potential, consistent with FAPs in mouse and human skeletal muscle.

Finally, we used the mouse lemur muscle single-cell suspensions to screen additional fluorophore-conjugated antibodies commonly used for the isolation of cell populations from mouse or human skeletal muscle. Using flow cytometry, we observed distinct populations of mouse lemur cells that stain for CD29 and CD82 (both used for human MuSCs), CXCR4 (used for mouse MuSCs), and CD34 (used for mouse and human FAPs, and mouse MuSCs), but not others such as CD45, LYVE1 or CD14 (used for human and mouse macrophages) and CD31 or CD48 (both used for human and mouse endothelial cells)

**Fig. 1 | Identification of myogenic and mesenchymal stem cell/progenitor populations from mouse lemur skeletal muscle. a** Brightfield images of mouse lemur myogenic cells in proliferation (top panel) or differentiation (bottom panel) medium. Myogenic cells were obtained by pre-plating on plastic and grown on collagen-coated dishes. **b, c** UMAP plot of 10X single cell RNAseq data. Colors reflect the two replicates (**b**) or the different cell types (**c**). **d** UMAP plot as in (**b, c**), but now with MYF5-positive cells (top) or PDGFRA-positive cells (bottom) highlighted. **e** UMAP plot as in (**b**), but with NCAM1-positive cells (top) or THY1-positive cells (bottom) highlighted. **f** FACS plots for anti-NCAM1 and anti-THY1 staining (top) or no antibody control (bottom). **g** Muscle cryosections from indicated species were stained for PAX7 and counterstained for DAPI and WGA. Arrowheads mark PAX7+ nuclei in the MuSC position under the basal lamina. **h** Muscle cryosections from mouse, mouse lemur, and humans were stained for PDGFRα and counterstained for DAPI and WGA. Arrowheads mark interstitial PDGFRα+ cells. **i** Cells were grown from a single NCAM1+THY1− cell isolated by FACS. Cells were grown in high serum (top) or in low serum (bottom). **j** Myoclone-derived myotubes

were stained for MYH2 protein. **k** NCAM1−THY1+ cells were expanded for 7 days before treatment with indicated differentiation media. Cells were stained with Oil Red O, Wheat Germ Agglutinin, or Alizarin Red, respectively. Insets are magnified views of areas noted by the boxes. **l** Bioluminescence imaging (BLI) post-transplantation. Presumptive MuSCs (NCAM1+THY1−) were purified by FACS, transduced with a lentivirus expressing GFP and Luciferase, and transplanted into the TA muscles of NSG recipient mice. n = 17 from 3 donors. Mice were imaged over time and normalized BLI signal was plotted as the mean plus standard error. **m** BLI after secondary transplantation. Engrafted MuSCs following transplantation (as in (**l**)) were purified by FACS and transplanted into the TA muscles of an irradiated NSG mouse. n = 3 from 3 donors. MuSC transplantation was followed by bioluminescence imaging and compared to control mice (irradiated, injured, but not transplanted). Image of BLI is shown on top. Data are presented as mean values ± SEM. The p-value was calculated by unpaired, two-tailed Student's t-test (*p < 0.05, p = 0.0388). Source data are provided as a Source Data file.

(Supplementary Fig. 1m, n, Supplementary Data 1). These antibodies will enable the purification and study of additional cell populations from mouse lemur tissues.

### Mouse lemur MuSCs engraft and self-renew in transplantation assays

We next asked whether the purified mouse lemur MuSCs could engraft in vivo. We transduced the FACS-purified NCAM1+/THY1− MuSCs from three aged mouse lemurs with a lentivirus expressing GFP and Luciferase and transplanted 10,000 cells into regenerating *tibialis anterior* (TA) muscles of immunocompromised Nod/Scid/Gamma (NSG) mice. We monitored engraftment by bioluminescence imaging (BLI) and observed an initial peak in signal followed by a plateau (Fig. 1l), consistent with BLI reported for mouse and human MuSCs[21–23]. We conclude that the mouse lemur MuSCs can engraft in skeletal muscle in vivo.

To confirm whether the engrafted cells are bona fide stem cells and capable of self-renewal to form new stem cells in vivo, we dissected the transplanted TA muscles and purified the MuSCs by FACS using GFP expression and NCAM1 staining (Supplementary Fig. 1o). We performed secondary transplantations with 1000 of these cells into irradiated, injured TA muscles of NSG mice again. We monitored engraftment by BLI and observed clear bioluminescence signal in transplanted legs after 21 days (Fig. 1m). Histological analysis revealed Luciferase-positive myofibers (Supplementary Fig. 1p). Moreover, we observed GFP-positive mononuclear cells in the satellite cell position (Supplementary Fig. 1q), confirming that the MuSCs had engrafted into the recipient muscle and contributed to new myofiber formation. We conclude that NCAM1+/THY1− cells are true MuSCs that can engraft, differentiate, and self-renew.

### Primate MuSCs take longer to break quiescence

A careful analysis of the lemur MuSC engraftment signal (Fig. 1l) showed that the BLI pattern differed from what we previously reported for mouse MuSCs[23]. For mouse lemur MuSCs, the signal increased more gradually. This could indicate differences in MuSC activation and expansion, although of course, we cannot exclude the possibility that the host muscle simply offered a suboptimal environment for mouse lemur MuSC expansion (which is not the case for mouse lemur FAPs, which engraft better than mouse FAPs (see below)). Nevertheless, we looked at EdU incorporation data from ex vivo MuSC activation time courses and observed that MuSCs from aged mouse lemurs take on average 12 h longer to enter the cell cycle compared to MuSCs from aged mice (Fig. 2a, Supplementary Fig. 2a, b). MuSCs from old humans showed EdU incorporation patterns comparable to MuSCs from aged mouse lemurs. Nevertheless, as estimated from the comparable slopes of the EdU incorporation curves, subsequent cell divisions occur at similar rates for all three species (Fig. 2a, Supplementary Fig. 2a, b). The

canonical marker for MuSC activation, MyoD[16,24], followed a similar pattern. Whereas ~75% of mouse MuSCs expressed MyoD protein after 36 h in culture, fewer mouse lemur or human MuSCs exhibited MyoD protein expression at this time point (Fig. 2b, c). The canonical marker for myogenic differentiation[25], MyoG, was expressed in ~50% mouse MuSCs after 4 days in culture, at which point only a few sporadic mouse lemur MuSCs showed nuclear MyoG protein staining (Supplementary Fig. 2c, d). We conclude that primate MuSCs take significantly longer to exit the quiescent state than mouse MuSCs, but still follow the canonical myogenic gene expression program.

### MyoD can drive mouse lemur MuSC activation and is down-regulated in primate MuSCs

We were interested in the molecular mechanisms that could explain these phenotypes. To this end, we performed cross-species comparisons of gene expression profiles. We were especially interested in comparing the transcriptomes of cell types identified in mouse lemur skeletal muscle with the transcriptomes of the corresponding cell types obtained from human[15] and mouse muscle[14,26]. We examined all genes that have been annotated in all three species and assigned as 1-to-1-to-1 orthologs by NCBI and Ensembl. We performed cross-species comparisons of gene expression levels on 11 cell types present in the three species to identify genes that are differentially expressed among the three species (see the "Methods" section, Supplementary Data 3)[27]. Initial comparisons revealed that the aged human MuSCs are molecularly more similar to aged mouse lemur MuSCs than to aged mouse MuSCs, something we study in greater detail in a separate publication[27]. We next plotted the main components of the NOTCH signaling pathway, which is among the best-studied mechanisms for maintaining MuSC quiescence in vivo[28,29]. Primate MuSCs showed higher levels of RNA expression of NOTCH target genes *HES1* and *HEY1* (Supplementary Fig. 2e). These data are consistent with a stronger quiescence phenotype of primate MuSCs in vivo. We then looked at known MuSC activation genes (Supplementary Fig. 2e). Primate MuSCs expressed lower levels of *MyoD1* RNA compared to mouse MuSCs, which we could confirm by microfluidic PCR (Supplementary Fig. 2f). Indeed, whereas up to 5% of freshly isolated mouse MuSCs stained positive for MyoD protein, consistent with prior reports[16], we could not find any MyoD-positive freshly isolated primate MuSCs (Supplementary Fig. 2g). In contrast, we could detect rare MyoD-positive cells among freshly isolated mouse MuSCs by single-cell western and in published CyTOF data (Supplementary Fig. 2h, i)[30]. To test whether MyoD can drive mouse lemur MuSC activation, similar to its role in mouse MuSCs[16], we overexpressed recombinant mouse MyoD in freshly isolated mouse lemur MuSCs. This caused increased MyoD protein expression and increased MuSC activation after 36 h (Supplementary Fig. 2j, k). Moreover, MyoD overexpression could induce increased levels of ID3 mRNA, a known

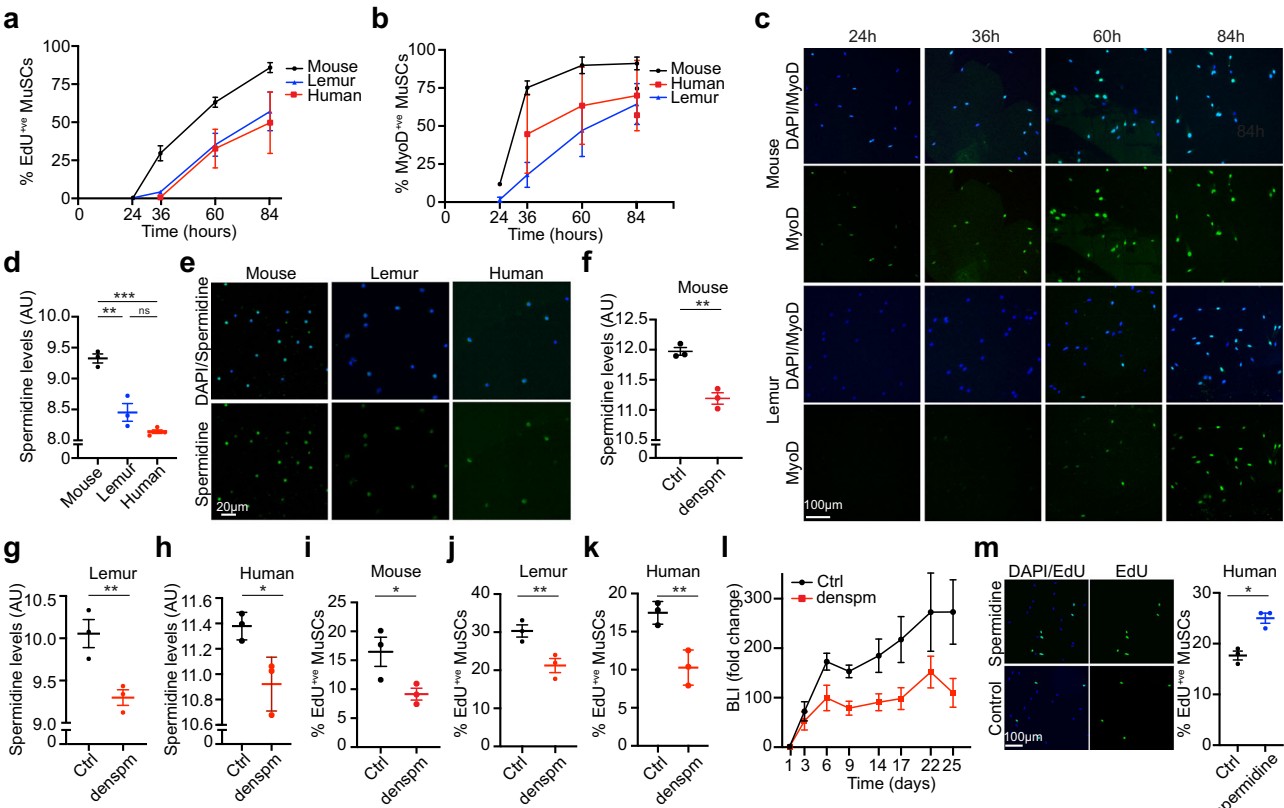

**Fig. 2 | Delayed quiescence exit in primate MuSCs. a** Time course of ex vivo EdU incorporation. *n* = 3. **b**, **c** Time course of MyoD protein expression during MuSC activation. Quantification is shown in (**b**), and representative images are shown in (**c**). *n* = 3 biological independent samples. **d**, **e** Freshly isolated MuSCs were stained for spermidine. Quantification is shown in (**d**) and representative images shown in (**e**). *n* = 3 biological independent samples. The *p*-values in (**d**) were calculated by unpaired, two-tailed Student's *t*-test. ***p* < 0.001 (*p* < 0.0001), ***p* < 0.01 (*p* = 0.0055), ns = *p* > 0.05 (*p* = 0.0537). **f**–**h** Proliferating MuSCs from mouse (**f**), *n* = 3 biological independent samples, mouse lemur (**g**), *n* = 3 biological independent samples, and human (**h**), *n* = 3 biological independent samples, were treated with SAT1 potentiator DENSPM or vehicle (Ctrl) for 24 h and stained for spermidine. **i**–**k** Proliferating MuSCs from mouse (**i**), *n* = 3 biological independent samples, mouse lemur (**j**), *n* = 3 biological independent samples, and human (**k**), *n* = 3 biological independent samples, were treated with SAT1 potentiator DENSPM or vehicle (Ctrl) for 24 h in the presence of EdU and stained for EdU. **l** Freshly isolated mouse lemur MuSCs were transduced with luciferase-expressing lentivirus and treated with either the SRM potentiator DENSPM or vehicle (Ctrl). Cells were transplanted into the TA muscles of recipient mice. Transplanted legs received an additional intramuscular dose of 20 µl 2.5 mM DENSPM on days 3 and 6. Engraftment was monitored by BLI. *n* = 6 from 2 donors. **m** Freshly isolated human MuSCs were treated with 100 nM spermidine or vehicle in the presence of EdU. Quantification is shown on the right, and representative images are shown on the left. *n* = 3 biological independent samples. All statistical data in Fig. 2 are presented as mean values ± SEM. All *p*-values in **f**–**k**, **m** were calculated by paired, one-tailed Student's *t*-tests. ***p* < 0.001, ***p* < 0.01, **p* < 0.05, ns = *p* > 0.05. The actual *p*-values were: **f** *p* = 0.009, **g** *p* = 0.005, **h** *p* = 0.0459, **i** *p* = 0.0214, **j** *p* = 0.0011, **k** *p* = 0.0025, **m** *p* = 0.0266. Source data are provided as a Source Data file.

MyoD target gene in myoblasts[31], in proliferating mouse lemur primary myoblasts (Supplementary Fig. 2l). These data suggest that MyoD has a similar role in mouse lemur myogenesis as it has in human and mouse myogenesis.

### Reduced spermidine levels contribute to the diminished MuSC expansion in vivo

Although lower MyoD expression levels can contribute to slower activation[16], we wanted to identify previously unrecognized molecular pathways that can affect MuSC activation. We therefore performed GO-term clustering analyses of differentially expressed genes in primate MuSCs compared to mouse MuSCs (Supplementary Fig. 2m, Supplementary Data 4). We found enriched terms related to mRNA splicing, mRNA translation, and various signaling pathways. One interesting GO term in these analyses was "Polyamine Biosynthetic Process" (Supplementary Fig. 2m). The polyamine synthesis pathway connects with many metabolic processes in the cell (Supplementary Fig. 2n)[32]. Some components of the polyamine biosynthesis pathway are early onset genes in MuSCs[33], suggesting they could be important

in the proliferative phase of the MuSCs. In our RNAseq data, we found that *OAZ1* and *SAT1*, whose activity would result in lower levels of the polyamine spermidine, were higher in primate MuSCs, while *AZIN1*, whose activity results in higher spermidine levels, was lower (Supplementary Fig. 2o). *SRM*, the rate-limiting step of spermidine biosynthesis, was also lower in primate MuSCs (Supplementary Fig. 2o). We confirmed these findings at the protein level (Supplementary Fig. 2p–r). Consistently, we found reduced steady-state levels of spermidine in primate MuSCs compared to aged mouse MuSCs (Fig. 2d, e).

We next tested the effects of spermidine breakdown on MuSC proliferation, using DENSPM[34], a small molecule potentiator of SAT1, the primary enzyme to break down spermidine and a negative regulator of hair follicle stem cell expansion[35]. DENSPM reduced spermidine levels in proliferating primary MuSCs, resulting in reduced EdU incorporation for all three species (Fig. 2f–k). We next tested this in vivo by transducing MuSCs from two mouse lemurs with luciferase-expressing lentivirus. Half of the cells were simultaneously treated with DENSPM to stimulate the breakdown of spermidine. We transplanted

these cells into the TA muscles of six immunocompromised mice and monitored engraftment by BLI. The transplanted muscles received an additional intramuscular injection of DENSPM or vehicle on days 3 and 6. As we anticipated, the muscles transplanted with DENSPM-treated MuSCs showed reduced BLI signal, consistent with reduced MuSC expansion in vivo (Fig. 2l). Finally, we treated human MuSCs with spermidine to directly increase spermidine levels. Spermidine treatment resulted in increased EdU activation (Fig. 2m). We conclude that spermidine levels are lower in primate MuSCs compared to mouse MuSCs, which contributes to the delayed activation phenotypes of primate MuSCs.

## A cell-intrinsic bias towards adipogenesis in primate FAPs

In parallel with our MuSC studies, we asked whether the purified mouse lemur FAPs could engraft in vivo. As with our MuSC studies, we transduced the FACS-purified THY1⁺/NCAM1⁻ FAPs from aged mouse lemurs with a lentivirus expressing GFP and Luciferase and transplanted 10,000 cells into regenerating TA muscles of NSG mice. We followed engraftment by BLI (Fig. 3a). We observed an initial increase in signal reflecting the engraftment and expansion of the FAPs. This signal reached a plateau that persisted for at least four weeks. In contrast, we did not see a persistent bioluminescence signal after transplantation of mouse FAPs (Supplementary Fig. 3a), which is

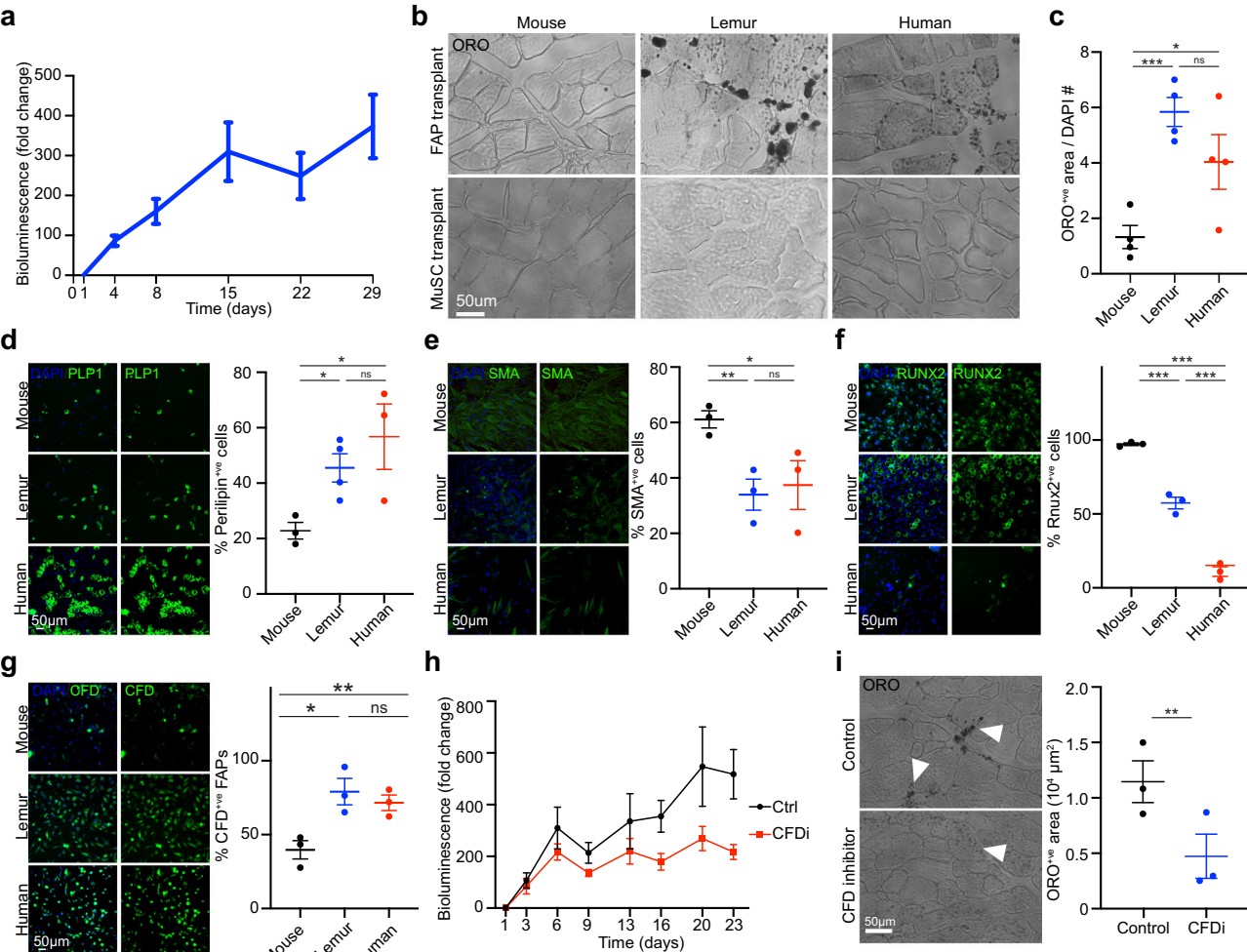

**Fig. 3 | CFD regulates the adipogenic fate of mouse lemur FAPs. a** BLI post-transplantation. Presumptive FAPs (NCAM1⁻THY1⁺) were purified by FACS from three mouse lemurs, transduced with a lentivirus-expressing GFP and Luciferase, and transplanted into the TA muscles of NSG recipient mice. Mice were imaged over time and the normalized BLI signal was plotted as the mean plus standard error. $n = 17$ from 3 donors. **b** Freshly isolated FAPs (top panels) or MuSCs (bottom panels) from mice, lemur, or humans were transplanted into the TA muscles of NSG mice. One month post-transplantation, muscles were harvested, fixed, and frozen. Cryosections were stained with ORO. **c** Quantification of ORO stains shown in (**b**). $n = 4$ biological independent samples. The $p$-values were calculated by unpaired, two-tailed Student's $t$-tests. ***$p < 0.001$ (p = 0.0005), *$p < 0.05$ ($p = 0.0445$), ns = $p > 0.05$ (0.1573). **d**–**f** Freshly isolated FAPs were expanded in vitro and treated with differentiation media for adipogenic (**d**), fibrogenic (**e**), or osteogenic (**f**) conversion. Cells were stained for Perilipin-1 (**d**), SMA (**e**), or RUNX2 (**f**). Left panels show representative images, right panels show graphs of quantification. $n = 3$ biological independent samples. **g** Freshly isolated FAPs were stained for CFD protein levels. Left panels show representative images, right panel shows the graph of

quantifications ($n = 3$ biological independent samples). **h** Freshly isolated FAPs were transduced with luciferase-expressing lentivirus. Half of the cells were simultaneously treated with the CFD inhibitor Danicopan or vehicle (Ctrl). Cells were transplanted into the TA muscles of three recipient NSG mice. Engraftment was monitored by BLI. $n = 6$ from 2 donors. **i** Transplanted muscles from **h** were isolated, fixed, and stained for ORO. Left panels show representative images with arrowheads denoting ORO staining, right panels show graphs of quantification. $n = 3$ biological independent samples. All statistical data in this figure are presented as mean values ± SEM. The $p$-values in **d**–**g** were calculated by unpaired, one-tailed Student's $t$-test and marked as ***$p < 0.001$, **$p < 0.01$, *$p < 0.05$, ns = $p > 0.05$. The actual $p$-values were: **d** mouse vs. lemur $p = 0.018$, mouse vs. human $p = 0.0247$, lemur vs. human $p = 0.1611$; **e** mouse vs. lemur $p = 0.0067$, mouse vs. human $p = 0.032$, lemur vs. human $p = 0.3781$; **f** mouse vs. lemur $p = 0.0003$, mouse vs. human $p = 0.0001$, lemur vs. human $p = 0.0006$; **g** mouse vs. lemur $p = 0.0111$, mouse vs. human $p = 0.0085$, lemur vs. human $p = 0.2536$. The $p$-value in **i** was calculated by paired, two-tailed Student's $t$-test (**$p < 0.01$, $p = 0.0072$). Source data are provided as a Source Data file.

consistent with published reports of mouse FAPs[36,37]. Because mouse and mouse lemur FAPs have multilineage potential (Fig. 1k, Supplementary Fig. 1k)[19,36,37], and because mouse FAPs have been shown to engraft in an adipogenic environment[36], we tested for the adipogenic fate of the translated mouse lemur FAPs. Indeed, when we stained the transplanted muscles with Oil Red O (ORO) and Luciferase antibodies, we observed clear interstitial staining in TA muscles that were transplanted with aged mouse lemur FAPs but not with aged mouse FAPs or aged MuSCs (Fig. 3b, c, Supplementary Fig. 3b). We also observed ORO-positive interstitial signal in TA muscles transplanted with old human FAPs (Fig. 3b, c). These data demonstrate that primate FAPs can form fatty infiltrates after transplantation.

Fatty infiltrates (myosteatosis) are common in aging human muscle and in muscular dystrophies and are thought to contribute to muscle pathology[38]. They are challenging to study because they are rare in aged wild-type C57 mice, the most commonly used mouse model[39]. However, the mouse lemur 10X single cell RNAseq revealed a tiny population of pre-adipocytes, marked by marker genes like *ADIPOQ* and *CIDEA* (Supplementary Fig. 3c). As such, we were interested in exploring the mouse lemur data for mechanisms that enable primate FAPs to form fatty infiltrates. We first confirmed the presence of fatty infiltrates in aged mouse lemur and human muscle sections, and their absence in aged mouse muscle sections (Supplementary Fig. 3d). Primate FAPs show a higher propensity to differentiate into ORO-positive cells in adipogenic medium, compared to mouse FAPs (Supplementary Fig. 3e). In contrast, compared to mouse FAPs, primate FAPs show decreased propensity to differentiate into fibroblasts or osteoblasts in fibrogenic or osteogenic medium, respectively (Supplementary Fig. 3f, g). We confirmed these results with cross-reacting antibodies against established late-stage differentiation markers. We observed higher levels of the adipocyte marker Perilipin1 in primate FAPs compared to mouse FAPs following adipogenic differentiation (Fig. 3d). In contrast, we observed lower levels of the fibrotic marker Smooth Muscle Actin in primate FAPs compared to mouse FAPs following fibrogenic differentiation (Fig. 3e). Finally, we observed lower levels of the osteocyte markers RUNX2 and Osterix in primate FAPs compared to mouse FAPs following osteogenic differentiation (Fig. 3f, Supplementary Fig. 3h). These data suggest that primate FAPs are biased towards an adipogenic fate.

To identify a molecular mechanism underlying the adipogenic predisposition of aged primate FAPs, we next analyzed our transcriptomic datasets. When we looked at the top 10 most significant differentially expressed genes between primate and mouse FAPs, we noticed complement factor D (*CFD*, Supplementary Fig. 3i, Supplementary Data 4). Although named for its role in the complement cascade, CFD can function as an adipokine to push mesenchymal stem cells toward adipogenesis[40,41]. Its function is unexplored in skeletal muscle. *CFD* expression was higher in many cell types in primate skeletal muscle, but it was particularly high in FAPs (Supplementary Fig. 3i). We could confirm higher levels of CFD expression at the protein level in primate compared to mouse FAPs using a CFD-specific antibody (Fig. 3g). We, therefore, tested the role of CFD in FAP adipogenesis using a small molecule inhibitor that blocks CFD activity. In vitro supplementation of the inhibitor limited the number of ORO-positive cells that differentiated from mouse lemur FAPs (Supplementary Fig. 3j). Strikingly, the inhibitor had no effect on mouse FAPs, which express much lower levels of CFD (Supplementary Fig. 3k). We confirmed these results with small interfering RNA against CFD, which caused reduced CFD protein levels and reduced adipogenesis in mouse lemur FAPs relative to control (Supplementary Fig. 3l, m). To test the role of CFD in vivo, we transduced FAPs that were freshly isolated from two mouse lemurs with a luciferase-expressing lentivirus. Half of the FAPs were simultaneously treated with CFD inhibitors. We transplanted these cells into the TA muscles of six immunocompromised mice and monitored engraftment by BLI. We observed a reduction in signal for the inhibitor-treated cells compared to control cells (Fig. 3h). Histologically, the transplanted TA muscles showed reduced ORO-positive fatty infiltration in the legs transplanted with inhibitor-treated cells compared to control cells (Fig. 3i). We conclude that signaling via CFD biases primate FAPs towards adipogenesis and that mouse lemur FAPs can be a good model to study dysregulation of human FAPs in aging and disease.

## Divergence among primate stem cells

One of the best established non-human primate models is the crab-eating macaque (*M. fascicularis*, also called long-tailed macaque or cynomolgus monkey)[42]. Although important due to its genetic proximity and similar physiology to humans, the macaque still has drawbacks as an animal model in terms of its relatively large size (<9 kg)[43] and relatively long lifespan (40 years)[44]. As such, we were interested to assess how the smaller (<90 g)[45] and shorter-lived (12 years)[45] mouse lemurs compared to macaques. To this end, we used publicly available, yet unannotated, single-cell RNAseq data generated from the skeletal muscles of two young (4-year-old) crab-eating macaques[46]. Following in-depth manual annotation of cell types, we integrated the data with our single-cell RNAseq data from human, mouse, and mouse lemur (Fig. 4a, b). The single-cell transcriptomes obtained from macaque muscle clustered together by cell type with the respective single-cell transcriptomes obtained from human, mouse, and mouse lemur, suggesting that mouse lemur could be as good a model organism as macaque. In fact, mouse lemur muscles do not contain two cell populations that are uniquely present in the macaque data (Fig. 4a, b). One population is enriched for pericyte and smooth muscle markers yet distinguishes itself from those cell types by high levels of CD44 and THY1 (Supplementary Data 5), indicating a progenitor-like cell type. Such a cell type has been described in the human aorta wall and has been suggested to indicate vascular neogenesis[47]. Importantly, CD44⁺/THY1⁺ pericytes can also be found in single nuclei RNAseq data obtained from macaque skeletal muscle[48]. Interestingly, the second macaque-specific cell population is a subset of capillary endothelial cells, the canonical interaction partner of pericytes (Fig. 4c, d). These capillary endothelial cells distinguish themselves from the other conserved capillary endothelial cells by expressing the adrenomedullin receptor cofactor RAMP3, which regulates angiogenesis (Supplementary Fig. 4a, b, Supplementary Data 5)[49]. Although needing functional validation, these molecularly defined cell types suggest divergence in the muscle vasculature between macaque and the other two primates and that mouse lemur is the non-human primate model that more closely resembles humans where it comes to the cell type composition of skeletal muscle. Moreover, these findings highlight the potential of using cross-species single-cell transcriptomic analyses to make informed decisions on the selection of non-human primate models and on which model best captures specific aspects of human biology.

We were next interested to compare gene expression patterns across all four species. We therefore calculated the gene expression levels for all genes with an annotated analog in all four species (Supplementary Data 6). When we plotted the top 10 genes that are expressed in MuSCs from all three primates but not in MuSCs from mice, we observed splice factor heterogeneous nuclear ribonucleoprotein A1 (*HNRNPA1*), the gene that is mutated in inclusion body myopathy and amyotrophic lateral sclerosis[50], as the top gene (Fig. 4e). We confirmed this at the protein level in muscle cryosections using an antibody raised against a conserved epitope in HNRNPA1, finding that in contrast to mouse MuSCs, primate MuSCs have high levels of HNRNPA1 protein staining (Fig. 4f, g). In contrast, we could detect robust expression of PAX7 in all four species (Fig. 4h). Finally, we could confirm higher expression of HNRNPA1 protein in freshly isolated human and mouse lemur MuSCs compared to mouse MuSCs (Supplementary Fig. 4c).

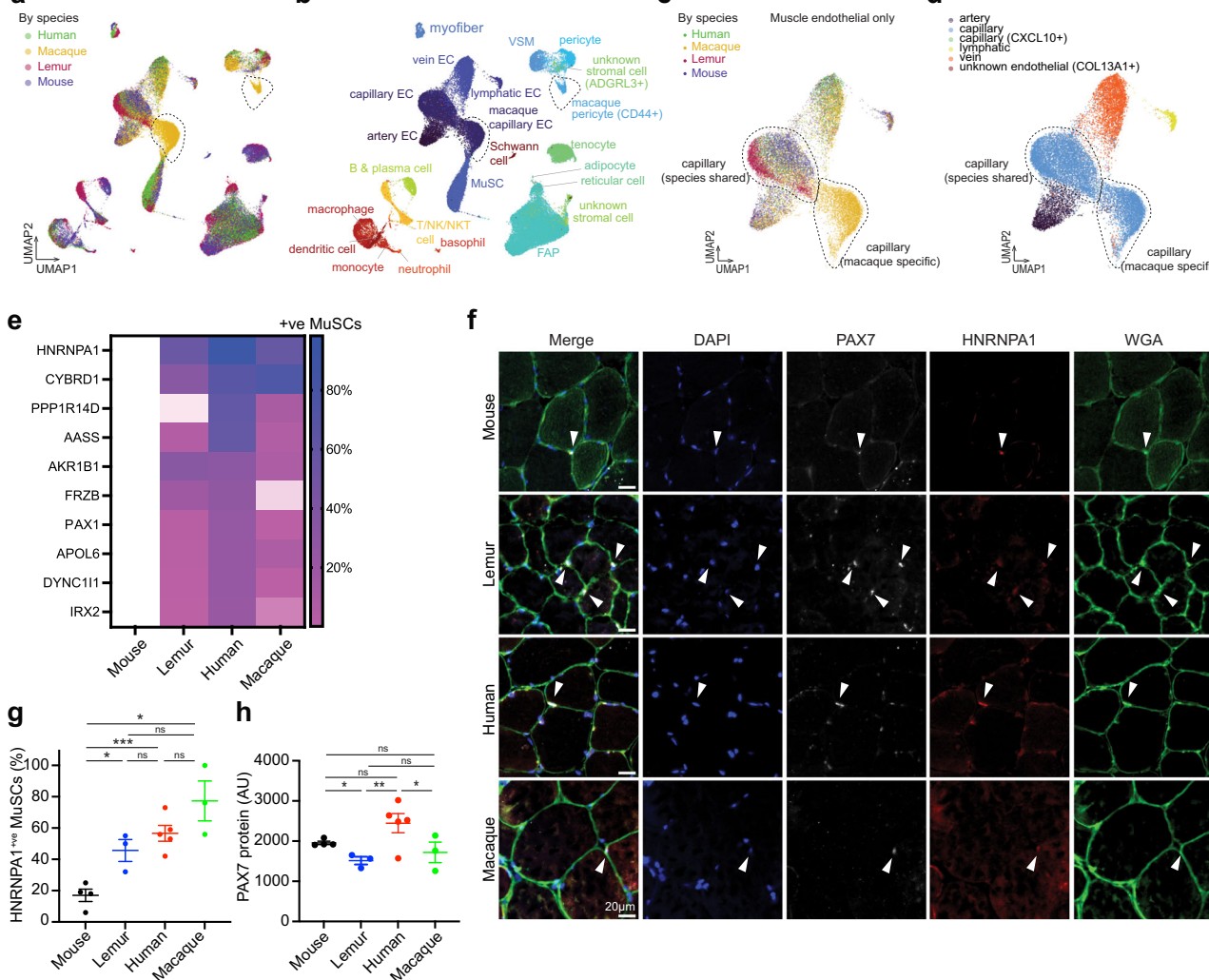

**Fig. 4 | Divergence of primate skeletal muscle stem cells. a, b** UMAP plot of 10X single cell RNAseq data. Colors reflect the four species (**a**) or the different cell types (**b**). Two cell clusters are specific to macaque, marked with dotted circles. **c, d** UMAP plot of 10X single-cell RNAseq data of the endothelial compartment only. Colors reflect the four species (**c**) or the different cell types (**d**). **e** Heatmap of the top 10 genes that are expressed in primate MuSCs but not in mouse MuSCs. **f–h** Muscle cryosections of indicated species were stained for HNRNPA1 and PAX7 and counterstained for DAPI and WGA. Arrowheads denote PAX7⁺HNRNPA1⁺ cells (**f**). Graphs show quantification of HNRNPA1⁺ MuSCs (**g**), and PAX7 staining intensity per MuSC (**h**). Statistical data in **g** and **h** are presented as mean values ± SEM.

Sample sizes for **g** and **h** are: mouse $n = 4$ biological independent samples, lemur $n = 3$ biological independent samples, human $n = 5$ biological independent samples, macaque $n = 3$ biological independent samples. The $p$-values in **g** and **h** were calculated by unpaired, one-tailed Welch's $t$-test and marked as ***$p < 0.001$, **$p < 0.01$, *$p < 0.05$, ns = $p > 0.05$. The actual $p$-values were: **g** mouse vs. lemur $p = 0.0163$, mouse vs. human $p = 0.0002$, lemur vs. human $p = 0.1357$, lemur vs. macaque $p = 0.057$, human vs. macaque $p = 0.1193$, mouse vs. macaque $p = 0.0161$; **h** mouse vs. lemur $p = 0.0147$, mouse vs. human $p = 0.0546$, lemur vs. human $p = 0.0072$, lemur vs. macaque $p = 0.2578$, human vs. macaque $p = 0.0454$, mouse vs. macaque $p = 0.2252$. Source data are provided as a Source Data file.

While most genes show similar expression patterns in stem cells across the three primates, this is not the case for all genes. We were interested to see if there were genes expressed in human stem cells that were similarly expressed in stem cells of one of the three model organisms, but not of the other two. To this end, we compared the MuSCs and FAPs from all four species and plotted genes that showed similar expression levels in MuSCs and/or FAPs from human and macaque or from human and mouse lemur, but different from mouse and mouse lemur or from mouse and macaque, respectively (Supplementary Fig. 4d–g). Glis Family Zinc Finger Protein 3 (*GLIS3*), which supports differentiation of embryonic stem cells[51], and the cell surface protein Leukemia Inhibitory Factor receptor (*LIFR*), which is key for adipogenic differentiation in embryonic stem cells[52], are higher in mouse lemur and human MuSCs and FAPs compared to mouse and macaque MuSCs and FAPs. This suggests a bias towards adipogenesis

in human and mouse lemur FAPs compared to macaque FAPs, which would be consistent with the apparent lack of fatty infiltration in published data on macaque muscle[53]. We could confirm that a larger fraction of mouse lemur and human MuSCs express GLIS3 protein in histology, compared to mouse and macaque MuSCs (Supplementary Fig. 4h, i).

## Muscle disease genes in primate stem cells

The identification of HNRNPA1 expression in primates but not mouse MuSCs led us to ask whether mouse lemurs could be a good model for muscle diseases. We therefore examined the expression patterns of genes that when mutated in humans cause a muscle disease. The single-cell nature of our data enabled us to ask whether any muscle disease genes are expressed in different cell types in primate muscle compared to mouse muscle, which could potentially result in

differences in phenotypes arising from mutations in that gene, between the different species. Toward this end, we compiled a list of 696 disease genes with a reported muscle phenotype (see the "Methods" section, Supplementary Data 7). Many disease genes have been experimentally mapped to cell type(s) of action[54–56]. Through manual curation, we assigned a "cell type of action" to all 696 disease genes, half of which mapped to myogenic cells. These annotations largely agreed (76%) with a recent muscle nuclei RNAseq study that assigned a "cell type of action" to muscle disease genes, based on single nucleus gene expression data combined with gene expression networks and metadata[56]. For those genes for which an annotated ortholog exists in all three species (566, 81%), we calculated and plotted the gene expression levels for each cell type (Supplementary Data 7) in order to compare gene expression profiles across cell types and across species.

We noticed that 372 (66%) disease genes with annotated orthologs were expressed in MuSCs in all three species. Nevertheless, some genes do have different expression patterns across cell types. In addition to the genes *MyoD1* and *HNRNPA1*, which we could confirm at the protein level (Fig. 4f, g, Supplementary Figs. 2f, g, 4c, Supplementary Data 7), we found that genes encoding components of the dystrophin–glycoprotein complex (DGC) are expressed in MuSCs. The DGC links the contractile apparatus of mature myofibers to the extracellular matrix and mutations in most DGC genes result in myofiber degeneration and muscular dystrophy[57]. We observed higher expression of the Dystrophin gene *DMD* in primate myofibers compared to mouse myofibers (Supplementary Fig. 4j). *DMD* is also expressed in MuSCs in all three species. However, its DGC partner Sarcoglycan A (*SGCA*), mutated in limb girdle muscular dystrophy R3, was higher in primate MuSCs compared to mouse MuSCs. Consistently, DGC components *SGCB*, *SGCD*, and *SGCE* are all expressed at higher levels in primates compared to mouse cells. We could confirm higher levels of SGCA expression in purified primate MuSCs at the protein level (Supplementary Fig. 4k), suggesting a potential role for SGCA in primate MuSCs. We conclude that muscle disease genes can be expressed in different cell types across species and that mouse lemur in general shows disease gene expression patterns across cell types that closely resemble human patterns.

### Primate genes without mouse orthologs

Finally, not all human genes have an ortholog in mice. Mouse lemurs could be a good model for the study of such primate-specific genes. The gene encoding the protease family with sequence similarity 111 member B (*FAM111B*), which is mutated in poikiloderma with tendon contractures and myopathy (OMIM 615704), was expressed at low levels in primate capillary endothelial cells (Supplementary Data 8). The gene encoding the lipolysis enzyme patatin-like phospholipase-domain containing 4 (*PNPLA4*), which is mutated in combined oxidative phosphorylation deficiency 1, was expressed in primate MuSCs and FAPs. The gene encoding myosin light chain 5 (*MYL5*), which lies in the disease locus for Merosin-positive Congenital Muscular Dystrophy (OMIM 609456)[58], was expressed in primate MuSCs and myofibers. MYL5 is important for spindle formation[59]. Finally, the Hes-family bHLH transcription factor 4 (*HES4*) is not disease-associated, but similarly unique to primates. HES4 acts downstream of NOTCH, can slow proliferation in neural stem cells, and is a non-redundant effector of human T-cell development[60,61]. In our transcriptomic data, HES4 was indeed expressed in T-cells, but also in MuSCs and pericytes. Mouse lemur may be a good model to study these primate-specific genes.

### Discussion

In this report, we employed single-cell RNA sequencing technologies in combination with flow cytometry to identify stem cell markers in the non-human primate gray mouse lemur model. Using these markers, we purified and functionally characterized two mouse lemur stem cell populations, MuSCs and FAPs, from skeletal muscle. Using cross-species analyses of single-cell gene expression profiles, we identified and validated differences in gene expression at the RNA and protein levels between primate and mouse cells. These expression differences had functional outcomes in the stem cells, identifying genes important for stem cell regulation. This pipeline is applicable to any model organism for which a reference genome exists.

Previous studies have reported on adult stem cells in the mouse lemur but have not been able to prospectively isolate them using cell surface markers. Differential plating has been used to study mesenchymal cells from lemur olfactory mucosa[62]. In addition, lemur neural stem cells have been identified by transcription factor staining and anatomical location[63]. In contrast, we describe the prospective isolation of non-human primate stem cells, allowing for the isolation of pure and well-defined populations. Our isolation is based on a panel of two markers, NCAM1 and THY1. The panel can be improved by additional negative selection for lineage markers. However, this will require the generation of mouse lemur-specific antibodies. Whereas MuSCs had been identified in non-human primates by immunostaining[64], prior attempts to use myogenic cells in transplantation experiments relied on myoblast-like cells obtained by selection using cell culture characteristics, and those cells exhibited limited engraftment[65,66]. It is well known that mouse MuSCs rapidly lose self-renewal potential ex vivo as they break the quiescent state[23], and this is consistent with the low engraftment potential of cells obtained after selection in vitro. The current identification of MuSCs in the primate mouse lemur may lead to improved transplantation models for non-human primates.

MuSCs exist primarily in a quiescent state of reversible cell cycle exit. Strikingly, our data show that both human and mouse lemur MuSCs take longer to exit the quiescent state than do mouse MuSCs. We cannot exclude the possibility that our assay conditions are suboptimal for primate MuSC activation. However, the differential expression levels in freshly isolated MuSCs of known quiescence regulators, including MyoD and NOTCH, suggest that the phenotype is cell-intrinsic.

Our molecular cross-species analysis revealed shared and divergent gene expression patterns. A comparative analysis of single-cell transcriptomes of the earliest stages of embryogenesis in human, marmoset, and mouse embryos similarly revealed common pluripotency factors as well as divergent signaling pathways such as DNMT1[67]. This highlights the value of molecular comparisons to identify the best non-human primate model. Open source data, such as transcriptomic atlases but also imaging repositories like the PRIMatE Data Exchange (PRIME-DE)[68], can help reduce the number of animals needed for experimentation.

One important degenerative tissue process for which current animal models are limited is that of age-related fatty infiltration in skeletal muscle. While we previously showed that mouse FAPs can be the source of fatty infiltrates in vivo[20], fatty infiltrates are not a prominent characteristic of aged mouse muscle. Through our comparative analysis, we uncovered a bias toward adipogenesis in primate FAPs and a potential cell-intrinsic mechanism controlling that cell fate decision. This not only highlights the potential for mouse lemur as a model for the study of age-related processes like fatty infiltration, but it also highlights the power of comparative single-cell transcriptomics combined with detailed phenotyping of a defined cell population. Interestingly, we identified adipocytes in the mouse lemur single-cell RNAseq data (Figs. 1c, 4b). Such pre-adipocytes were not present in the macaque data (Fig. 4a, b). Intriguingly, no incidents of ectopic fat in skeletal muscle were reported in a histological study of 660 crab-eating macaques[53]. Our datasets, in combination with the recent mouse and human cell atlas papers[14,15,26,27,69], enable the identification of unique cell surface markers for the purification of pure cell types for the downstream analysis of cell function.

The single-cell transcriptome integration revealed that mouse lemur muscle has a similar cell-type composition to humans. This is not

the case for the macaque muscle, which contains two unique molecular cell types not found in the muscles of humans, mouse lemurs, or mice: a pericyte progenitor cell and a capillary endothelial progenitor cell. This indicates cell type specification in the muscle vasculature. Interestingly, the macaque-specific cell types are not present in lung data from the same animals[27], suggesting they reflect a muscle-specific adaptation. Although we cannot exclude that these cell types are only a feature of young macaque muscles, we could identify the transcriptomic signature in published single nucleus RNAseq data from slightly older macaque muscles (6 years old), but not in our transcriptomic data from young mice (including 1 and 3-month old mice) and humans (including a 38-year old donor)[14,15,48]. Nevertheless, it will be important to integrate single-cell sequencing data from young mouse lemur and human muscle to confirm that the cell types truly are unique to macaque muscle and to validate their existence in histology using cross-reacting antibodies. It should however be noted that it has been possible to develop muscle-tropic adeno-associated virus capsids that target macaque and mouse muscle in vivo[70]. This suggests that in general there is good conservation between these species, as is also supported by our analyses where most endothelial and smooth muscle cells localize to clusters that are shared between all four species. Nevertheless, the absence of progenitor-like cells in mouse lemur suggests that mouse lemur may be a more suitable model for studies of skeletal muscle homeostasis and regeneration that involve vascular biology.

Finally, our data revealed that the expression pattern of disease genes across cell types is largely conserved across the three species, although the expression levels in each cell type are often more similar between the two primates than between mice and humans. Mouse lemur may be a good model to study these disease genes, especially in cases where the mouse model does not recapitulate the human pathology. The data might also point the way to potential paralogs that can take over protein function, in cases where a protein of known importance is found not to be expressed in one species. For example, primate myofibers express higher levels of *MYBPC1*, whereas mouse myofibers express higher levels of the paralog *MYBPC3*[71]. Similarly, Utrophin is important for myofiber integrity, yet mouse lemur myofibers do not express detectable *Utrophin* mRNA, suggesting there may be a paralog. In two mouse lemur cell atlas studies, we report divergent expression patterns for a large majority (89%) of conserved genes between human, macaque, mouse lemur, and mouse single-cell transcriptomes, suggesting selective pressures for the expression of protein-coding genes[27,72]. While generating targeted knockouts in mouse lemurs remains an open question for the scientific community, it is already possible to identify and study naturally occurring disease variants. By careful animal phenotyping with genome sequencing, naturally occurring null mutations in mouse lemur genes can be identified. One such study described a mouse lemur with neuromuscular phenotypes and naturally occurring, potentially pathogenic variants in the disease genes *DAG1*, *POMGNT2*, *LARGE2*, and *CACNA1A*[73]. This illustrates that naturally occurring variants can be identified after careful phenotyping of individual mouse lemurs. Such variants can shed new light on disease pathogenesis.

## Methods

### Animals
Gray mouse lemurs originated from the closed captive breeding colony at the Muséum National d'Histoire Naturelle in Brunoy, France, and were transferred to Stanford University and were maintained for non-invasive phenotyping and genetic research as approved by the Stanford University Administrative Panel on Laboratory Animal Care (APLAC #27439) and in accordance with the Guide for the Care and Use of Laboratory Animals[74]. Briefly, mouse lemurs were individually or in groups housed indoors in an AAALAC-accredited facility in a temperature (24 °C) and light-controlled environment (daily 14:10 h and

10:14 h light:dark alternating every 6 months to stimulate photoperiod-dependent breeding behavior and metabolic changes) with perches and nest boxes, and were fed fresh fruits and vegetables, crushed primate chow plus live insect larvae as enrichment items. Health and welfare were routinely monitored and clinical care was provided by the Veterinary Service Center. Animals in declining health that did not respond to standard therapy were euthanized by pentobarbital overdose under isoflurane anesthesia[74]. Prior to euthanasia, a veterinary examination was performed, and animal body weight was obtained. At the time of euthanasia, all lemurs had been living in summer-like long days (14:10 h) for at least 3 months (range 3-6 months), showing standard activity patterns without signs of torpor. Mouse lemurs analyzed in this study were a 10-year-old female (used for the initial 10X single cell RNAseq; corresponds to L2 in the Tabula Microcebus cell atlas[27]), an 11-year-old male (used for the SS2 experiments and cell purification studies; corresponds to L4 in the Tabula Microcebus cell atlas[27]), an 11-year-old male (used for the cell purification studies and 10X single cell RNAseq; L5), an 11-year-old female (used for the cell purification studies; L6), and a 13-year-old female (used for the cell purification studies; L7), each sacrificed for humane reasons at old age. All postmortem tissue analyses were approved by the Animal Welfare Board at Aarhus University.

Crab-eating macaque skeletal muscle biopsies were obtained from Boehringer-Ingelheim under a Materials Transfer Agreement. All biopsies had been obtained postmortem and opportunistically from control animals from other approved experiments. Samples analyzed here are from one 3-year-old male and two 3-year-old females. The muscle samples for RNA sequencing were obtained from a previously published dataset and contained data from one 4-year-old male and one 4-year-old female crab-eating macaque[46].

Human tissue biopsies were sourced from the Tabula Sapiens project through collaboration with a not-for-profit organization, Donor Network West (DNW) (San Ramon, CA, USA)[15]. DNW is a federally mandated organ procurement organization (OPO) for Northern California. Recovery of non-transplantable organs and tissues was considered for research studies only after obtaining records of first-person authorization (i.e., donor's consent during his/her DMV registrations) and/or consent from the family members of the donor. The research protocol was approved by the DNW's internal ethics committee (Research project STAN-19-104) and the medical advisory board, as well as by the Institutional Review Board at Stanford University which determined that this project does not meet the definition of human subject research as defined in federal regulations 45 CFR 46.102 or 21 CFR 50.3. Donors were a 59-year-old female, a 61-year-old female, and a 51-year-old male, corresponding to donors TSP1, TSP2, and TSP14 in the Tabula Sapiens cohort[15]. In addition, we sampled *latissimus dorsi* from a 54-year-old male and a 73-year-old male.

Mouse procedures were approved by the Administrative Panel on Laboratory Animal Care of the VA Palo Alto Health Care System as well as the Animal Welfare Inspectorate Denmark (Dyreforsøgstilsynet 2023-15-0201-01441). NSG mice (strain 005557, NOD.Cg-Prkdcscid Il2rgtm1Wjl/SzJ) were purchased from The Jackson Laboratory. Mice were housed in specific pathogen-free conditions in barrier-protected rooms under a 12-h light–dark cycle and were fed *ad libitum*. Male mice were used for analyses. All wild-type mouse studies were done with mice aged 28–36 months. All transplantations were done using NSG mice aged 3–6 months.

### MuSC growth and differentiation
Single cells were seeded in half-area 96-well plates (4680, Corning) coated with ECM (Gel from Engelbreth-Holm-Swarm murine sarcoma, E1270, Sigma, 1:100) and cultured in growth medium (Ham's F10 medium (11574436, Fisher Scientific) supplemented with 20% horse serum (26050088, Thermo Fisher), 1% Pen/Strep (15140122, Thermo Fisher), and FGF (2.5 ng/ml, 100-18B, Peprotech)). After 14 days, cells

were switched to myogenic differentiation medium (Ham's F10 medium supplemented with 2% horse serum, 1% Pen/Strep). After 14 more days, myotubes were imaged. To grow myoblasts, pooled cells were seeded in 12-well plates coated with collagen. After four days, cells were trypsinized (0.5% Trypsin–EDTA, 10779413, Fisher Scientific) and stained with anti-human NCAM1-PE-Cy7 for purification by FACS. Purified cells were seeded on collagen-coated dishes and maintained in proliferation medium (DMEM/F10 1;1, supplemented with 15% FBS, 1% Pen/Strep, and recombinant FGF (2.5 ng/ml)). The SAT1 potentiator DENSPM (0468, Tocris) was dissolved in DMSO and added to the cells at a final concentration of 2.5 mM. Spermidine (# S0266, Sigma-Aldrich) was used at 100 nM. For transfections, cells were treated with Lipofectamine 2000 (11668030, Invitrogen) and OptiMEM (31985070, Thermo Scientific) using the reverse transfection protocol according to the manufacturer's guidelines. Cells were washed with medium 12 h following transfection. Transfection was evaluated by GFP protein expression after 36 h.

### MuSC activation

MuSCs were seeded in half-area 96-well plates coated with ECM (1:100) and cultured in Wash Medium (Ham's F10 medium supplemented with 10% horse serum, 1% Pen/Strep). For EdU pulse-chase experiments, EdU was added at the time of plating, or, when the cells were transfected, EdU was added after 12 h when the transfection medium was replaced with Wash Medium. EdU was stained using the ClickIT EdU reaction kit (C10337, Thermo Scientific) according to the manufacturer's instructions.

### Mouse lemur myoblast culture and transfection

Mouse lemur primary myoblasts were grown in high serum medium (DMEM/F10 1;1, supplemented with 15% FBS, 1% Pen/Strep, and recombinant FGF (2.5 ng/ml)) on collagen-coated (C8919, Sigma) dishes. Collagen coating was performed as previously described[75]. For differentiation, cells were switched to a low serum differentiation medium (DMEM/F10 1;1, supplemented with 5% horse serum and 1% Pen/Strep). For transfections, proliferating mouse lemur myoblasts were seeded in collagen-coated six-well plates at 100,000 cells per well. After 2 h, myoblasts were treated with plasmid DNA expressing GFP or mouse MyoD dissolved in OptiMEM with Lipofectamine 2000 according to manufacturer's guidelines. At 24 h following transfection, GFP expression was confirmed by fluorescence microscopy and cells were lysed for RNA extractions.

### Quantitative RT-PCR

Total RNA from mouse lemur myoblasts was isolated by using RNeasy Micro Kit (74034, Qiagen). Isolated RNA was reverse transcribed using the High-Capacity cDNA Reverse Transcription Kit (4374967, Thermo Scientific) and qPCR was carried out on a LightCycler 480 system (Roche) using SYBR Green (A46109, Thermo Scientific), and gene-specific primers (Invitrogen). Relative quantification of gene expression was calculated by using the comparative CT method using HPRT as the reference gene. PCR primer pairs used were: lemur HPRTfw TGCCGAGGATTTGGAAAAGG; lemur HPRTrv GCCTCCCATCTCCTTC ATCA; lemur ID3fw CGGGAGAGGACTGTGAACTT; lemur ID3rv AAGGA GACCAGAAGACCAGC; lemur MyoDfw GCTCTGGGGTTCCTCTTCCTT; lemur MyoDrv CTAGGGGTGGGGCTTAAGTC; mouse HPRTfw CAGTA CAGCCCCAAAATGGTTA; mouse HPRTrv AGTCTGGCCTGTATCCAA CA.; mouse MyoDfw AGTGAATGAGGCCTTCGAGA; mouse MyoDrv CAGGATCTCCACCTTGGGTA.

### Microfluidic RT-PCR

Microfluidic PCR was performed according to Fluidigm protocols and as described previously[16,76,77]. Five hundred cells were sorted into 5 µl lysis buffer of the CellsDirect One-Step qRT-PCR Kit (11753500, Thermo Fisher). PCR mix and primers were added and cDNA

synthesized followed by preamplification for 20 cycles. Pre-amplified cDNA was diluted and loaded on a 48.48 dynamic array (BMK-M-48.48, Fluidigm), and analyzed with a BioMark HD. Calculations were performed with PCR software (Fluidigm).

### Single-cell western

Single-cell westerns were performed in accordance with the manufacturer's protocol (Milo SC-Western, Protein Simple) as previously reported[19,78]. Milo scWest chips were rehydrated in a Suspension Buffer for 30 min and loaded with single-cell suspensions of mouse MuSCs (100,000 cells/ml). Well occupancy was monitored by brightfield microscopy after ~10 min and chips were washed to remove cells not loaded in a well. Loaded chips were then inserted into the Milo System and run with the following parameters: 10 s lysis, and 60 s electrophoresis. Antibody probing was performed with antibodies to MyoD (5.8A, BD, 1;5, or NBP2-32882, Novus Biological, 1;20) and TUBB (ab6046, Abcam, 1:40 dilution). Stained and dried chips were scanned on a GenePix Microarray Scanner and images for each spectral channel were analyzed with Scout Software (Protein Simple). Only lanes positive for TUBB were used for quantification of 98 and 332 cells per replicate. MuSCs from MyoD$^{-/-}$ mice were used as a negative control.

### FAP growth and differentiation

Single cells were seeded in 96-well plates coated with ECM (1:100) and cultured in growth medium (Ham's F10 with 10% FBS (11573397, Fisher Scientific), 1% Pen/strep). For differentiation experiments, pooled cells were seeded in 12-well plates coated with ECM. Cells were switched to mesenchymal differentiation medium when they reached 90% confluency[19,20]. For fibrogenic differentiation, cells were maintained in fibrogenic medium (F10 with 10% FBS, 1% Pen/Strep, 1 ng/ml TGFβ1) for 3 weeks. For osteogenic differentiation, cells were maintained in osteogenic medium (F10 with 10% FBS, 1% Pen/Strep, 1 nM BMP-4) for three weeks. For adipogenic differentiation, cells were maintained in adipogenic medium (F10 with 10% FBS, 1% Pen/Strep, 0.25 µM dexamethasone (D4902, Sigma), 0.5 mM isobutylmethylxanthine (I5879, Sigma), 1 µg/ml insulin (I6634, Sigma), 5 µM troglitazone (A6355, Sigma)) for 2 weeks. For the CFD inhibitor experiments, the FAPs were maintained in high serum for 4 days and then switched to adipogenic medium. The CFD inhibitor Danicopan (MedChem Express, Cat no. HY-117930) was dissolved in DMSO (BP231, Fisher Scientific) and added to the adipogenic medium at a final concentration of 1 mM and replenished every 2 days for 2 weeks. For siRNA transfections (Silencer Select, Thermo Fisher (siCFD sense: GGAUAAGGGUGUCAGGUAAtt)), the FAPs were treated with lipofectamine-siRNA mixtures according to the manufacturer's protocol. FAPs were washed, incubated with a transfection mix in OptiMEM for 6 h. Subsequently, cells were washed and switched to adipogenic medium. Transfection was repeated on day 3.

### Myoclone assay

Half-area, flat bottom 96-well plates were coated overnight with Collagen and Laminin (23017-015, Invitrogen)[18]. Plates were washed 3 times with PBS and all wells excluding those on the edges were filled with 100 ml medium containing 20% horse serum, 1% Pen/Strep, and FGF (5 ng/ml)[18]. Single cells were sorted using a BD ARIA III FACS. Clonal cultures were maintained for 8 days and then stained with Hoechst dye and imaged using a wide-field microscope, or stained with an antibody against MYH2 and DAPI.

### Antibodies

We summarize all tested lemur antibodies and reagents in Supplementary Data 1. For immunofluorescence studies, we selected antibodies raised against conserved epitopes. In addition, the following antibodies were used in this study: anti-luciferase (Sigma-Aldrich, #L0159, 1:200); Streptavidin-647, anti-rabbit-biotin, anti-rabbit-647, anti-rabbit-488, anti-mouse-488 (all Invitrogen, 1:1000), WGA-488

(Thermo-Fisher, #W11261, 1;200), WGA-568Plus (Thermo-Fisher, #W56133, 1;200), anti-GFP (Invitrogen, #A11122, 1;100). To isolate mouse lemur stem cells, we used anti-CD56-PE-Cy7 (MEM-188), anti-THY1-APC (5E10) (all Biolegend, 1;100). To isolate human stem cells, we used anti-CD45-FITC (HI30), anti-CD31-APC (WM59), anti-THY1-PE (5E10), and anti-CD82-PE-Cy7 (ASL24) (all Biolegend, 1;100). To isolate mouse stem cells, we used anti-CD45-FITC (30-F11), anti-CD31-FITC (MEC13.3), anti-SCA1-PacBlue (D7), and anti-VCAM1-PE-Cy7 (429) (all Biolegend, 1;100).

## Immunofluorescence staining

Freshly isolated cells were cytospun and fixed with 4% PFA. Cells were washed 2 times with PBS and stored at 4 °C. For stainings, cells were permeabilized with 0.3% Triton x-100 in PBS for 5 min (or 10 min for the nuclear HNRNPA1). Cells were washed with PBS and blocked using 10% Donkey Serum for 30 min. Primary antibodies were incubated overnight at 4 °C in 10% Donkey Serum. The next day, cells were washed 3 times for five min each with 0.3% Triton. Secondary antibodies were added in 10% Donkey Serum and incubated for 1 h. Cells were washed 3 times for five min each with 0.3% Triton and incubated for 15 min with DAPI in PBS. Cells were washed 3 times with PBS and stored at 4 °C in the dark until imaging. Cells were imaged using a Zeiss Axiofluor inverted microscope with a cooled CCD camera and images were quantified using Volocity software or ImageJ-Fiji.

## Lentiviral transduction

Luciferase and GFP protein reporters were subcloned into a third-generation HIV-1 lentiviral vector (CD51X DPS, SystemBio)[23]. To transduce freshly isolated MuSCs, cells were plated at a density of 10,000 cells per well in a 24-well plate and incubated with 30 µl of concentrated virus and 6 µl polybrene (TR1003G, Fisher Scientific). Plates were spun for 60 min at $3200 \times g$ at 25 °C. Cells were then washed with fresh media two times, scraped from plates in 50 µl 1.2% BaCl$_2$.

## Transplantation

Recipient NSG mice were injected with 10,000 (primary) or 1000 (secondary transplant) purified MuSCs or FAPs resuspended in 50 µl 1.2% BaCl$_2$ (342920, Sigma). Live cells were counted with a haemocytometer prior to transplantation. The injection was performed with a Hamilton syringe. The 27-gauge needle was inserted into the proximal TA muscle and the cells were slowly injected into the muscle. The needle was left undisturbed for 5 min in the muscle before extraction. For the transplantations in the presence of the small molecules, the CFD inhibitor Danicopan was added to the cells at a final concentration of 1 mM, while DENSPM was added to the cells at a final concentration of 2.5 mM. At days 3 and 6 post-transplantation, the mice received an extra injection of the small molecule or vehicle in a volume of 20 µl.

## Irradiation

Before secondary transplantation, NSG mice were anesthetized with ketamine (95 mg/kg) and xylazine (8 mg/kg) by intraperitoneal injection. We then irradiated hindlimbs with a single 12 Gy dose, with the rest of the body shielded in a lead jig. We performed transplantations within 24 h of irradiation.

## Mouse lemur tissue dissociation and stem cell isolation

Skeletal muscles were harvested postmortem from the mouse lemurs by an experienced veterinarian. Tissues were mechanically dissociated to yield a fragmented muscle suspension. This was followed by a 30 min digestion in a Collagenase II (500 units per ml; LS004177, Worthington, Invitrogen) Ham's F10 solution, with 10% horse serum and 1% Pen/Strep. Samples were spun down at $1500 \times g$ for 5 min and the supernatant was removed. A second digestion was performed for 20 min with Collagenase II (100 units per ml) and Dispase (2 units per ml; 17105041, Thermo Scientific, Invitrogen), again in Ham's F10

medium with 10% horse serum and 1% Pen/Strep. The resulting cell suspensions were pulled through 25-gauge needles and pushed through 100 and 40 µm filters. The resulting single-cell suspensions were stained with antibodies for 15 min at 4°C with head-over-head rotation prior to FACS (anti-NCAM1-PE-Cy7 1:100) and anti-THY1-APC. Cells were washed, filtered, and analyzed on calibrated BD-FACS Aria II or BD FACSAria III flow cytometers equipped with 488, 633, and 405 nm lasers and FACSDIVA 8+ software (BD Biosciences) to obtain the MuSC and FAP populations. Mouse lemur MuSCs were NCAM1$^+$THY1$^-$, and mouse lemur FAPs were NCAM1$^-$THY1$^+$. Data were analyzed using FlowJo software (v10.8.1, BD Biosciences).

## Human and mouse tissue dissociation and stem cell isolation

Human abdominal muscle biopsies were harvested postmortem from abdominal muscle by an experienced surgeon as reported previously[15]. Mouse lower hindlimb muscles were harvested postmortem as described previously[75]. Tissues were mechanically dissociated and digested for 30 min in a Collagenase II (500 units per ml; Invitrogen) Ham's F10 solution, with 10% horse serum and 1% Pen/Strep. Samples were spun down at $1500 \times g$ for 5 min and the supernatant was removed. A second digestion was performed for 20 min with Collagenase II (100 units per ml) and Dispase (2 units per ml; Invitrogen), again in Ham's F10 medium with 10% horse serum and 1% Pen/Strep. The resulting cell suspensions were pulled through 25-gauge needles and pushed through 100 and 40 µm filters. The resulting single-cell suspensions were stained with antibodies for 15 min at 4 °C with head-over-head rotation prior to FACS. The human samples were stained with anti-CD82-PE-Cy7 (1:100), anti-THY1-PE (1:100), anti-CD45-FITC (1:100), and anti-CD31-APC (1:100). The mouse samples were stained with anti-VCAM1-PE-Cy7 (1:100), anti-Sca1-PacBlue (1:100), anti-CD45-FITC (1:100), and anti-CD31-FITC (1:100). Cells were washed, filtered, and analyzed on calibrated BD-FACS Aria II or BD FACSAria III flow cytometers equipped with 488, 633, and 405 nm lasers to obtain the MuSC and FAP populations. Human MuSCs were CD82$^+$THY1$^-$CD45$^-$CD31$^-$ and human FAPs were CD82$^-$THY1$^+$CD45$^-$CD31$^-$. Mouse MuSCs were VCAM1$^+$SCA1$^-$CD45$^-$CD31$^-$ and mouse FAPs were VCAM1$^-$SCA1$^+$CD45$^-$CD31$^-$.

## Bioluminescence imaging

Bioluminescence imaging was performed using the Xenogen IVIS-Spectrum System (Caliper Life Sciences). Mice were anaesthetized using 2% isoflurane at a flow rate of 2.5 l/min. Intraperitoneal injection of D-Luciferin (50 mg/ml, L-8220, Biosynth International Inc.) dissolved in sterile PBS was administered. Immediately following the injection, mice were imaged for 60 s at maximum sensitivity (f-stop 1) at the highest resolution (small binning). Every minute, a 60 s exposure was taken until the peak intensity of the bioluminescent signal began to diminish. Each image was saved for subsequent analysis. Imaging was performed in a blinded fashion; the investigators performing the imaging did not know the identity of the experimental conditions for the transplanted cells.

## Bioluminescence image analysis

Analysis of each image was performed using Living Image Software, version 4.0 (Caliper Life Sciences, Perkin Elmer). A manually generated circle was placed on top of the region of interest and resized to surround the limb or the specified region on the recipient mouse. Similarly, a background region of interest was placed on a region of a mouse outside the transplanted leg. The peak value for each limb was noted together with the background value at the corresponding time point. Bioluminescence data were calculated relative to the background and plotted with the first time point set to 1.

## Histology

TA muscles were carefully dissected away from the bone, embedded in Tissue Tek and frozen in cooled isopentane. For Oil Red O

assessments, TA muscles were fixed for 24 h in 1% PFA followed by 24 h in 20% sucrose. Tissues were washed in PBS, briefly dried on a tissue to remove excess fluid, embedded in OCT, and frozen in cooled iso-pentane. Frozen tissues were cryosectioned on a Leica CM3050S cryostat at a thickness of 10 μm, mounted on Fisherbrand Colorfrost slides, and stained. For colorimetric staining with H&E, samples were processed according to the manufacturer's recommended protocols.

## Immunohistology

Muscle cryosections were washed with PBS, stained with WGA, washed with PBS and fixed for 10 min in 10% formalin. Excess formalin was washed off with PBS and slides were subjected to antigen retrieval using a citrate buffer for 1 h. Slides were cooled down, blocked with 10% donkey serum, and stained overnight at 4 °C with primary anti-bodies (mouse anti-Pax7 1;100, rabbit anti-HNRNPA1 1;100, rabbit anti-PDGFRα 1;100, rabbit anti-GLIS3 1;100). The next morning, slides were washed and incubated with secondary antibodies for 1 h at room temperature. Slides were washed and mounted using DAPI-containing mounting medium. Stained slides were imaged immediately using an Olympus BX63 Upright Widefield Fluorescence microscope with cooled CCD camera.

## Cross-species quantifications of immunohistology

Tissues were imaged at ×20 magnification at fixed exposure settings for all species. Focusing and channel acquisition order was set starting with DAPI, as a focused configuration of nuclei is paramount for ana-lysis, with 0 μm offset for subsequent channels to register coplanar signal. PAX7 (alexa647) and HNRNPA1 (alexa488) or GLIS3 (alexa488) were imaged right after DAPI, and before WGA (alexa568Plus) as the last channel, so as to provide the least possible amount of photo-bleaching to the channels used for quantification (PAX7, HNRNPA1/GLIS3). Raw data images were analyzed using ImageJ-Fiji. Nuclei were identified using a DAPI mask, whereupon PAX7 signal was used to rank nuclei. Visual inspection enabled the selection of quantifiable nuclei that had to conform to the MuSC position characteristics (adjacent to myofiber) and have visually clear PAX7 staining. Rare interstitial cells were excluded from the analysis as they might reflect activated MuSCs. Cells in areas of high background (e.g. borders of the section) were excluded from the analysis.

## CyTOF analysis

Published single-cell mass cytometry (CyTOF) data were obtained from flowrepository.org (accession FR-FCM-ZY3F, FR-FCM-ZY3W) and analyzed using Cell Engine as previously described[30,79]. The experi-ments contained a time course of muscle injury; we only analyzed the data from uninjured muscles. Briefly, live cells were identified by DNA label and cisplatin (dead cell dye)[30]. Subsequently, cells were nega-tively gated for CD31 (MEC 13.3 conjugated to Sm154), CD45 (30-F11 conjugated to Sm147), and SCA1 (E13-161.7 conjugated to Nd142) to exclude the endothelial cells, immune cells and FAPs, and positively gained for ITGA7 (3C12 conjugated to Ho165) and PAX7 (PAX7 con-jugated to EU153) to select the MuSCs. The fraction of MyoD$^{+ve}$ MuSCs (5.8A conjugated to Dy164) was quantified.

## Single-cell RNAseq analysis and cross-species comparisons

Sequence data (scRNAseq gene expression Counts/UMI tables, and cellular metadata) are available on Figshare (https://figshare.com/projects/Tabula_Microcebus/112227). Data can be explored inter-actively using CellXGene on the Tabula Microcebus portal: https://tabulamicrocebus.ds.czbiohub.org/. Raw sequencing data and gen-ome alignments are available on request and described in detail in a preprint[27]. Standard procedures for filtering, variable gene selection, dimensionality reduction, and clustering were performed using the Seurat package version 2.2.1. Cells were annotated using Seurat or CellXGene based on known marker genes. For the cross-species

analysis, we used the 10X RNAseq profiles from *rectus abdominis* muscles from two 55+-year-old human donors[15] and the 10X RNAseq profiles from limb muscles from the old animals in the Tabula Muris Senis cohort (18, 21, 24, 30 months old)[14]. In addition, we integrated 10X RNAseq data from 2 young crab-eating macaques[46]. We combined cells from all donors and performed the cross-species analysis for all cell types that were represented by more than 20 cells in each of the three species and for genes that had an annotated paralog in all three species.

## Disease gene compilation

We searched the OMIM database for disorders with a reported muscle phenotype or pathology, using the gene table of neuromus-cular disorders[80] as a starting point and adding musculoskeletal and rhabdomyolysis disorders. For each gene, we list the main disease name (certain genes cause multiple diseases, in which case only one disease is named here), the disease group, and the gene function based on GeneCards descriptions, as well as the PubMed ID for the papers identifying the mutation and the mouse model. We grouped genes by the cell type in which we predict the genetic defect to manifest first, based on functional studies, gene expression reports (cell atlas data and protein atlas data), and phenotypic descriptions. We also included "cell of action" annotations by Eraslan et al.[56] We then plotted the gene expression levels for each gene across cell types and across species.

## Cross-species analyses of 10X single-cell RNAseq data

The methods for cross-species analyses are described in detail in a preprint[27]. We used published data obtained previously by us from human (1 female and 1 male, aged 45–65) and mouse (7 females and 10 males, aged 18–30 months) skeletal muscle[14,15]. These data had been acquired using the same methodologies as the mouse lemur data. Cell type annotations were manually re-annotated where necessary to achieve consistent naming with the mouse lemur annotations. Only cell types with at least 20 profiled cells in each of the three species were selected for use in the comparisons. Cell type similarity across species was calculated using Self-Assembling Manifold mapping (SAMap), a cross-species mapping method, using default parameters[81,82]. We next assembled a list of genes with annotated orthologs in all three species. To this end, we retrieved the corresponding human and mouse orthologs for all mouse lemur genes annotated in NCBI. Mouse lemur genes that did not have an assigned human and mouse ortholog, or that had multiple, were removed (15,297 out of 31,966 unique mouse lemur NCBI gene IDs). We next removed genes that did not have an ortholog listed in Ensembl and MGI. We performed differential gene expression using the Wilcoxon rank sum test, of lemur vs. mouse and human vs. mouse. Using a log fold-change threshold of 10 and a *p*-value threshold of $10^{-5}$, we identified significant genes shared by both comparisons. For the Gene Ontology analyses, we used longer gene sets derived from a log fold-change threshold of 2 and a *p*-value of $10^{-5}$. Finally, we appended the one-to-one orthologs between human and crab-eating macaque, as assigned by Ensembl. A total of ~15,000 one-to-one gene orthologs were thus uncovered across human, lemur, and mouse, and ~13,000 across human, lemur, mouse and crab-eating macaque. Sequence identity was based on those reported in the Ensembl homology database. Gene lists were uploaded in DAVID ver-sion 2021 (https://david.ncifcrf.gov/) and Functional Annotation Clustering was performed at default settings including GO term analysis.

To look for primate-specific muscle disease genes (i.e., orthologs exist in human and mouse lemur but are absent in mouse genomes), we looked for human genes that have an ortholog gene in mouse lemur but have no ortholog in mouse, according to ortholog assignments by NCBI and Ensembl. Four such genes were identified and examined in this study. Because they are not among the 1-to-1-to-1 orthologs, we

normalized their expression to the total transcripts of the cell (not limited to the 1-to-1-to-1 orthologs) when visualizing their expression.

To identify genes with selective expression in human and lemur vs. mouse or in human and lemur vs. mouse and macaque, or in human and macaque, vs. lemur and mouse, we performed two-tailed Wilcoxon rank-sum tests comparing expression independently for each gene and each homologous cell type as well as tailed Wilcoxon rank-sum tests comparing expression in the cells of the cell type vs. all rest cells in the dataset. Next, for each cell type, we searched for three categories of genes: First, genes with significantly primate-enriched expression, which requires (1) cell type average expression of the gene is above 0.5 in both primates and (2) 5-fold greater expression and $p < 1e-5$ when comparing both the human and lemur cell types vs. the homologous mouse cell type. Second, genes with significantly lower expression in primate cells, which requires (1) cell type average expression of the gene is above 0.5 in mouse, and (2) 5-fold lower expression and $p < 1e-5$ when comparing both the human and lemur cell types vs. the homologous mouse cell type. Third, genes that are significantly enriched in the cell type compared to the other cell types, regardless of the species, which requires (1) cell type average expression of the gene is above 0.5 in all three species, and (2) 5-fold greater expression and $p < 1e-5$ when comparing this cell type vs. other cell types.

## Statistical analysis and reproducibility

Data points represent biological replicates and are graphed with an error bar showing the SEM. Groups were analyzed in unpaired experiments using two-tailed Student's $t$-tests. When in specific follow-up experiments the null hypothesis clearly included differences in one direction, one-tailed Student's $t$-tests were used. When the same sample was split into two conditions, paired $t$-tests were used. A Welch correction was performed when sample size was unequal. All data points shown reflect biologically independent replicates. Experiments were done on a single lemur sample whenever a lemur sample became available. Experiments in Fig. 1a, h and Supplementary Figs. 1g, j and 3d were performed once with qualitative images shown. Experiments in Fig. 1g, i, j, k and Supplementary Figs. 1f, I, k, l, p, q, and 3b were performed at least 3 times, with qualitative images shown. Data were compiled using Microsoft Excel. Graphs were rendered and statistics calculated using Prism 10 (Graphpad). All data are presented as means + SEM, ns = $p > 0.05$, *$p \leq 0.05$, **$p \leq 0.01$, ***$p \leq 0.001$.

## Reporting summary

Further information on research design is available in the Nature Portfolio Reporting Summary linked to this article.

## Data availability

Tabula Microcebus mouse lemur scRNAseq gene expression counts/ UMI tables and cellular metadata used in this study are available on figshare [https://figshare.com/projects/Tabula_Microcebus/112227][27]. For the cross-species comparison, human data were from the 10X data of the "Tabula Sapiens for the muscle" [https://figshare.com/projects/Tabula_Sapiens/100973][15]. Mouse data were from 10X data of the "Tabula Muris Senis" [https://figshare.com/articles/dataset/Processed_files_to_use_with_scanpy_/8273102/2][14]. Crab-eating macaque data were from the 10X data of the cynomolgus monkey cell atlas, available on Zenodo [https://zenodo.org/records/5881495#.ZERMCnbMKUk][46]. All other data supporting the findings of this study are available within the article and its supplementary files. Any additional requests for information can be directed to, and will be fulfilled by, the corresponding authors. Source data are provided with this paper.

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

## Acknowledgements

We thank the Palo Alto Veterans Affairs Flow Cytometry Core and the FACS Core Facility, Aarhus University, Denmark for assistance with flow cytometry experiments. We thank the Bioimaging Core Facility, Health, Aarhus University, Denmark, for the use of equipment and support. We thank Dr. Gabe Guerrero for assistance with processing mouse lemur muscles. Macaque muscle specimens were acquired through an MTA with Dr. Antonio Filareto and Suzanne Segal at Boehringer-Ingelheim. We thank Dr. Eric Jabart at Protein Simple for assistance in performing the single-cell westerns. This work was generously supported by the Chan-Zuckerberg Foundation, as well as grants from Novo Nordisk Foundation (Start Package 0071116), Danmarks Frie Forskningsfond (3101-00380B), and Aarhus University Foundation (AUFF-E-2022-9-28) to A.D.M. and grants from the NIH (R01 AG068667, R01 AR073248, and P01 AG036695) to T.A.R.

## Author contributions

A.D.M. initiated the project. A.D.M., J.K., A.K., M.P.J.S., Z.F., J.F., A.U., A.S.C., H.I., S.B., S.K., E.P. performed experimental analyses. A.D.M., S.L., C.E., and O.B. performed bioinformatics analyses. SL., C.E., and M.K. provided the mouse lemur samples and coordinated single-cell RNA sequencing. T.A.R. provided guidance throughout. A.D.M. and T.A.R. wrote the manuscript.

## Competing interests
The authors declare no competing interests.
