## [Transparent Peer Review file · Nature Communications]

In vivo self-renewal and expansion of quiescent stem cells from a non-human primate

Corresponding Author: Dr Thomas Rando

Version 0:

Reviewer comments:

Reviewer #1

(Remarks to the Author)

In this manuscript, the authors performed single cell RNAseq and fluorescence-activated cell sorting in the muscle of non-human primate mouse lemur and compared the transcriptional profile and functional properties of MuSCs and FAPs with those from mouse and human muscles. Data indicate that mouse lemur MuSCs and FAPs are more similar to human than to mouse counterparts.

This study is mostly descriptive and of comparative nature, with very limited novelty. The conclusions are quite confirmatory over what has been already known in mouse and human models, and it is unclear what significant conceptual advance this manuscript provides in term of MuSC and FAP biology. Furthermore, the interpretation of the data is often complicated by lack of specific reagents and conclusive information.

This reviewer appreciates the extensive work made on a new animal model, with the purpose to show that it can represent a model closer to humans and therefore more reliable than the mouse model; however, the conclusion that non-human primates represent a model closer to human than mice was highly predictable. This reviewer also acknowledges that the analysis presented have some potential to reveal differences in skeletal muscle properties across animals; yet, they remains underdeveloped at this stage and therefore are not very informative.

This manuscript seems definitely more appropriate for a more specialized journal.

Below are some specific points that this review recommend to address to improve the quality of this work, regardless of where this manuscript will be published.

- 1) There are intrinsic technical limitations that reduce the interest for this work. For instance, the advantage of using this model is limited by the fact that studies are restricted to quiescent cells from post-mortem animals.
- 2) scRNAseq samples from the two animals used are not homogeneous in term of numbers of single cell isolated and analysed.
- 3) Data comparison with publicly available single cell RNAseq data generated from skeletal muscles from two young (4-year old) crab-eating macaques are problematic in term of age differences with other models
- 4) The conclusions on comparative analysis are purely speculative, although of potential interest, without a functional follow-up. Perhaps, the unicity of macaque specific cell populations could turn into an interesting point to develop, as it might reveal specie-specific muscle properties to exploit for future therapeutic purposes. However, this requires experimental evidence that are not presented in this manuscript.
- 5) There are many genes that appear not expressed/detected in some specie, despite their function has been well established and essential for muscle development and homeostasis. It would be interesting to determine whether functional paralogs are expressed that mediate similar functions across species.
- 6) The authors mention that some antibodies work because of their binding to conserved epitopes, but nowhere in the manuscript they provide protein alignments for the three different species
- 7) The conclusion that overexpression of MYOD in lemur satellite cells leads to activation of the cells is based on the

increased expression of MYOD in MYOD transfected cells compared to GFP control. This is not surprising and should be corroborated by analysis of downstream transcriptional targets of MYOD

8) It would be important to show also the % of cells expressing protein involved in the spermidine pathway and not only the total intensity of IF

Reviewer #2

(Remarks to the Author)

This is an interesting manuscript that delves into characteristics of lemur mouse muscle stem and progenitors as well as mesenchymal (fibro-adipogenic cells). It describes various comparisons between lemur muscle cells to those of the mice and human counterparts and makes interesting conclusions about the suitability and limitations of each model organism and its relevance in the human context.

I have the following concerns regarding this manuscript:

Low number of Pax7 expressing cells in the lemur scRNA-seq data is not well explained or justified. Most of the analysis regarding scRNA-seq is global clustering. It would be interesting to perform cell-specific clustering to reveal heterogeneity in lemur, mouse and human muscle stem and FAPs cells.

The FACS strategy in the paper is unclear and needs to be better described and demonstrated. From this reviewer's understanding, as shown in extended Figure 1E, it appears that the lemur muscle stem cells are only identified and isolated with CD56 (NCAM1), which is a known marker used in human MuSC isolation. However, only using a single antibody is not appropriate for isolating a pure population of cells. Several publications isolate human muscle stem cells, using negative markers (CD11b, CD31, CD34, CD45) along with 1 or 2 positive markers (NCAM1, CXCR4, and CD82). This is also seen in the FAPs isolation, the authors appear to be indicating that the FAPs are isolated using only a THY1 antibody, which beyond the issues identified above of only using one antibody for isolating a pure population, THY1 is a well known marker of fibroblasts.

Negative markers (CD31, CD45 and SCA1) are used in FACS experiments testing the use of the known positive marker VCAM-1 (extended Figure 1A). However, it does not appear that they were used afterwards. This may explain the poor purity results that the authors obtained from their sorting of lemur MuSCs, from RNA-Seq only 80% expressed markers associated with MuSCs, while when attempting to culture the cells, only 20% were capable of clonal growth. With similar results seen in the FAPs. The authors must repeat these experiments using a proper FACS antibody panel, including negative selection markers and preferably a second positive marker.

The authors claim that the human specific antibodies typically used in MuSC FACS did not stain the Lemur muscle stem cells, however no explanation was given as to the cause. It would be important to investigate the epitope that these antibodies recognize and determine whether that region is conserved between human and lemurs. Further, the authors should investigate whether the antibodies used to sort mouse MuSCs would be appropriate for use in the lemur context.

Without being able to clearly demonstrate the isolation of a pure population of cells, subsequent experiments may be in question as any differences observed could simply be attributed to the fact that the cells isolated from the lemur are not pure MuSCs nor FAPs.

In order to determine whether the cells sorted were functional muscle stem cells, the authors used lentiviral infection to have the cells express luciferase and GFP. Afterwards the cells were transplanted into a host mouse, then retransplanted into another host mouse. While the authors see an increase in BLI intensity after transplantation and that persists after retransplantation, these results are not conclusive. All they indicate is that cells derived from lemur muscle can populate and persist in mouse muscle. There is no evidence that these are muscle stem cells. It would be essential for the authors to take cross sections of the transplanted muscle and stain for Pax7 and Laminin, to determine whether the transplanted cells are able to home to the muscle stem cell niche and populate the correct anatomical location. The presence of GFP+/PAX7+ cells under the laminin of a myofiber would be indicative of muscle stem cells. The presence of luciferase positive myofibers only indicate that some of the cells were at the very least myogenic progenitors, but it is not conclusive that they were muscle stem cells. As it stands, the results from the BLI experiment are open to interpretation.

The luciferase staining of the transplanted muscle cross section is not convincing (extended Figure 1N). The signal may simply be background, it is necessary to show at least one negative myofiber as it defies expectation that every single myofiber regenerated using the transplanted cells, and to the same degree to have the same level of luciferase, considering there was only 1000 cells transplanted. The likelihood that these luciferase positive myofibers are simply due to background is supported by the cross section staining in extended Figure 3L, where luciferase expressing FAPs were transplanted into the muscle. Yet we see that every myofiber is positive for luciferase, despite the fact that FAPs do not fuse with myofibers and therefore the myofibers cannot have obtained luciferase expression.

The growth media used in this paper for MuSC/myoblast growth is F10 supplemented with 20% HS, 1% P/S and 2.5 ng/mL of FGF. To this reviewer's knowledge there is no published paper using this specific media for the growth of MuSCs or myoblasts in culture. Did the authors mean 20% FBS instead, and if not what is the justification for using 20% HS. Please show the individual channels for all staining pictures. Particularly the MyoD staining for cultured cells and the TA cross sections.

Can the authors clarify why they performed a 14 day differentiation assay of the myoblasts when in humans 6-8 days is the

typical time to fully differentiate and in mice 5-7 days is sufficient.

In extended Figure 3L, the legend states that these are TA cross sections of NSG mice that had MuSCs transplanted in them. However, in the text it says that this experiment was done using transplanted FAPs. Please correct this.

The authors write "To test whether MyoD can drive mouse lemur MuSC activation, similar to its role in mouse MuSCs, we overexpressed recombinant mouse MyoD in freshly isolated mouse lemur MuSCs. This caused increased MyoD protein expression and increased MuSC activation after 24h (Extended Data Fig. 2f,g)." This statement does not reflect the experiment performed as EDU was added for 36 hours, not 24 hours, and the experimental design is also unclear. From this reviewer's understanding, muscle stem cells were isolated and then transfected with either MyoD or GFP using lipofectamine, transfection itself takes between 1-3 days to be able to see overexpression of the protein of interest. Therefore, I do not understand whether the EDU was added during the transfection step, thereby causing the EDU and the transfection to occur simultaneously, which would not be ideal as the cells would not be over expressing MyoD initially and would be growing in transfection reagent. Or was the EDU added after transfection was completed, in which case this experiment cannot be assessing the role of MyoD in activation as the cells would have already activated by the time the EDU was added. Further, the increase in EDU positive cells does not appear to be biologically significant as the control condition sees approximately 0.6% positive cells, while the MyoD O/E has about 1.2% positive. This is an increase that does not come near the levels seen in the WT mouse MuSCs at 36 hours (~25% EDU positive), therefore the claim that mouse MuSCs activate faster due to their increase MyoD levels is not supported by this experiment. Lastly, there are no representative staining pictures which are necessary to validate the claims made by the authors.

The materials and methods sections are missing some important details. For examples, on page 15, line 563 the authors state "Single cells were seeded in 96-well plates coated with ECM (1:100)" which ECM? Catalogue number? Page 15, line 582, the author state "For fibrogenic differentiation, cells were 582 maintained in fibrogenic medium (10% FBS, 1% Pen/Strep, 1 ng/ml TGFβ1)" what's the base medium? Is it DMEM?, F10? Something else?

Reviewer #3

(Remarks to the Author)

In this manuscript, Morree et al. report the identification and functional characterization of two quiescent stem cell populations (MuSCs and FAPs) in the non-human primate *Microcebus murinus*. The authors utilized single-cell RNA sequencing in combination with flow cytometry to identify NCAM1 and THY1 as stem cell markers in the gray mouse lemur, a novel model organism. Using these markers, they purified and functionally characterized MuSCs and FAPs from skeletal muscle. Additionally, they identified and validated differences in gene expression at both the RNA and protein levels between primate and mouse cells.

In general, the research is engaging. The majority of findings are well-organized and trustworthy, aligning with the standards of "Nature Communications". Nevertheless, there is potential for further enhancement if feasible. More information can be found below:

1. Lines 80-81 describe that the authors analyzed 3,122 single cells from one sample and 9,409 single cells from another sample in their single-cell sequencing (Fig. 1b). How to explain the significant difference between these two samples?
2. Line 86-88, the authors described antibodies used for purification of mouse MuSCs were not effective for labeling lemur MuSCs (Extended Data Fig. 1a, b). Consider using humanized antibodies to retry this assay.
3. Line 170-176, the authors concluded that NCAM1+/THY- cells are true MuSCs that can engraft, differentiate, and self-renew (Fig.1l). How about the function of newly formed myofiber? Please make it clear.
4. Line 190-193, the authors concluded fewer mouse lemur or human MuSCs exhibited MyoD protein expression during MuSC activation (Fig.2b). The results of immunostaining alone are not sufficient to support this conclusion. Please include western blot or mass spectrometry experiments to strengthen these findings.
5. Line 246-253, the authors indicated that lower spermidine levels in primate contributed to delayed activation phenotypes of MuSCs (Fig.2j). Would adding spermidine to MuSCs in primates restore the delayed phenotype?
6. Line 213-214, the authors described that primate MuSCs showed higher levels of RNA expression of NOTCH target genes HES1 and HEY1 (Extended Data Fig. 2c). From the graph, we can see that HES1 expression is extremely high in the mouse lemur, while HEY1 expression is comparable across all three species. Please explain it.
7. Line 216-217, the authors concluded that Primate MuSCs expressed lower levels of MyoD1 RNA compared to mouse MuSCs. Please confirm this result by quantitative real-time PCR.
8. Line 235-239, the authors indicated that protein levels of OAZ1, SAT1 and SRM are higher in primate MuSCs compared with mouse (Extended Data Fig. 2k-m). There is an inconsistency between the microscopic images and the statistical data. Please include western blot or mass spectrometry experiments to enhance the reliability of these results.
9. In Fig.3d-3g, the immunostaining results do not sufficiently demonstrate the differences in adipocyte markers among the three species due to notable discrepancies between the microscopic images and the statistical data. Please incorporate quantitative experiments, such as mass spectrometry, to clarify these differences.
10. Line 302-315, the authors evaluated CFD's role in FAP adipogenesis using a small molecule inhibitor to block CFD activity. Due to potential negative effects of the inhibitor on cells, generating CFD knockout or knockdown cell lines would provide a clearer understanding of CFD's role both in vitro and in vivo.
11. Line 323-330 conclude that there are two unique cell populations present in macaques (Fig. 4a-b). However, the single-cell sequencing data for macaques come from young individuals, while data for mice, humans, and lemurs are from older individuals. Could the presence of these two cell populations in macaques be attributable to age differences? Please clarify this point.

12. Line 402-405 describe higher levels of SGCA expression in purified primate MuSCs at the protein level (Extended Data Fig. 4i). However, the immunostaining images for mice and lemurs do not show noticeable green fluorescence. Please include western blot or mass spectrometry experiments to validate this conclusion.

13. Line 413-420, the authors described FAM111B is expressed at low levels in primate capillary endothelial cells and MYL5 is expressed in primate MuSCs and myofibers (Extended Data Fig. 4j). However, the expression of FAM111B and MYL5 was not observed in the primate cells shown in the bubble graph. Please clarify this discrepancy.

Version 1:

Reviewer comments:

Reviewer #1

(Remarks to the Author)

The authors have fairly addressed most of the reviewer comments and the revised version of the manuscript is definitely improved

Reviewer #2

(Remarks to the Author)

The revised manuscript shows many improvements, however, there are still some issues of clarity and the need for further refinement.

Major concerns:

1. The one major issue remaining is the isolation of the cells of interest (MuSCs and FAPs) from the lemurs. This reviewer is still not convinced on the purity of the cells isolated, mainly due to the FACS strategy. Doing a single positive selection with no negative selection raises doubts on the identity of the sorted cells. While I could be convinced on the NCAM1 sorting for the MuSCs as it is not a common marker, mostly seen in neurons and it is highly expressed in MuSCs and myogenic cells. However, that is not the case for THY1. While it is expressed in FAPs, it is also expressed in a multitude of other cell types, many of which are present in the muscle, including fibroblasts, adipocytes. The confirmatory staining of Pdgfra is not particularly useful in this case as fibroblasts can express Pdgfra as well. The purity of the lemur MuSC and FAPs isolation should also be compared to the purity the authors see from both the mouse and human samples. Until the authors can fully validate the cells they are using, no solid conclusion can be derived from all subsequent experiments.

2. The TA PAX7 staining looks fine, but the claim in the manuscript is that these cells are in a sublaminal position "Furthermore, we observed PAX7+ cells in the MuSC-position under the basal lamina in mouse lemur muscle cryosections", yet no laminin staining was done, therefore the authors cannot claim that these are PAX7 positive cells in the proper anatomical location. To fully confirm that this PAX7 staining is labeling MuSCs there must be a laminin staining as well.

There is a consistent issue throughout the paper, anytime a cross-section staining for PAX7 or GFP (in the case of the BLI transplantation experiment), the cross-section has only WGA counterstaining to mark cell membranes and the WGA staining is not particularly clean either. However, that is not sufficient to show that these are MuSCs in the proper location as the positive cells could easily be in the interstitial space.

The transplantation staining of GFP cells should ideally be redone with laminin and PAX7 to properly show whether these GFP cells are in the correct anatomical location and are expressing the MuSC marker PAX7.

Minor concerns:

1. After FACS sorting the MuSCs, they are stained for Pax7, and FAPs were stained for Pdgfra. While these cells were cytopun onto a slide, it is not stated in the manuscript, only in the methods. It would be clearer to say that freshly isolated MuSCs and FAPs were stained. Further, no quantification is given for the percent Pax7+ or Pdgfra positive cells. It simply states in the manuscript that they are positive. However, we see in the representative image that the FAPs are not 100% positive, nor could the MuSCs be 100% positive.

2. The FAPs TA cross section staining with PDGFRA is unconvincing. There is far too much non-specific staining to make any conclusion. Another antibody must be used.

3. In the DENSPM treatment it should be made clear in the results section that the mice were given multiple doses of the compound after transplantation. Otherwise, it makes it seem as if the cells were only treated prior to transplantation. Also, the concentration used should be added to the figure legend or the figure itself and not just in the methods section (Figure 2L)

4. Representative image for EDU staining of spermidine-treated cells (Figure 2M) and the concentration of spermidine added should be in the figure or the legend

Reviewer #3

(Remarks to the Author)

I have reviewed the revised manuscript and the authors have satisfactorily addressed all the previous comments. The changes made improve the clarity and quality of the manuscript. I believe the paper is now suitable for publication in Nature Communications.

I have no further comments and recommend accepting the manuscript for publication.

Version 2:

Reviewer comments:

Reviewer #2

(Remarks to the Author)

The response of the authors to my previous comments and the modifications they have made in the revised manuscript are reasonable and I have no further comments. I think the manuscript is now suited for publication.

RESPONSE TO REVIEWERS' COMMENTS

GENERAL RESPONSES

We thank the Reviewers for their careful reading of the manuscript and for their constructive comments to improve the manuscript overall. Below we have responded in a point-by-point fashion to each comment by each Reviewer. We have made several major revisions to the manuscript in response to Reviewer suggestions:

- We included new images to better reflect the mean image quantifications. Images were deconvoluted and the background was subtracted.
- We added extensive immunohistological analyses to complement the cell biological and transcriptomic analyses. This includes extended characterization of the muscle histology following transplantation, testing by FACS of 23 additional antibody clones against the human and mouse paralogs of common cell surface markers, including CD31 and CD45, additional analyses of purified MuSCs, and additional antibody stains across all four species.
- We added new molecular analyses, including microfluidic PCRs and single cell westerns as well as in silico analyses of published single cell transcriptomic datasets.

SPECIFIC REVISIONS

Among the additional studies and findings that are now included in the revised manuscript that respond directly to comments/criticisms of one or more Reviewers and that further strengthen our manuscript are the following:

- Fig. 1J: MYH2 staining on myoclone-derived mouse lemur myotubes. These data support our model that the mouse lemur MuSCs give rise to mature myotubes.
- Fig. 2M: Analysis of human MuSCs treated with spermidine or control. These data support our model that spermidine levels regulate primate MuSC quiescence.
- Supplementary Fig. 1J: MYH2 staining on bulk MuSCs-derived mouse lemur myotubes. These data support our model that the mouse lemur MuSCs give rise to mature myotubes.
- Supplementary Fig. 1L: MyoD staining on purified mouse lemur FAPs cultured in low serum. These data support our findings that the purified FAPs fractions are not contaminated with myogenic cells.
- Supplementary Fig. 1P: Extended images of luciferase-positive and luciferase-negative myofibers after mouse lemur MuSC transplantation. These data demonstrate that only the regenerating myofibers are positive for the MuSC lineage-tracer.

- Supplementary Fig. 1Q: GFP staining of the secondary transplant muscles, revealing the presence of rare GFP-positive mononuclear cells in the satellite cell position under the basal lamina. These data further support the model that mouse lemur MuSCs can generate quiescent progeny that occupy the stem cell niche.
- Supplementary Fig. 2A: Gating strategies for obtaining mouse MuSCs and FAPs.
- Supplementary Fig. 2B: Gating strategies for obtaining human MuSCs and FAPs.
- Supplementary Fig. 2F: Microfluidic quantitative qRT-PCR analyses of purified MuSCs from mice and mouse lemurs. These data confirm that mouse MuSCs express high levels of MyoD mRNA, whereas mouse lemur MuSCs do not. This is consistent with our model of the role of MyoD in MuSC quiescence and activation.
- Supplementary Fig. 2H: Single cell western assays for mouse MuSCs demonstrate that a small subset of mouse MuSCs expresses the MyoD protein. The data support our model that MyoD is important for MuSC activation.
- Supplementary Fig. 2I: Analysis of mouse muscle single cell suspensions by CyTOF, revealing that a small subset of mouse MuSCs express the MyoD protein.
- Supplementary Fig. 2L: Quantitative qRT-PCR analyses of primary mouse lemur myoblasts transfected with MyoD or control plasmid. These data show that mouse MyoD increases the expression of its canonical target gene ID3. This is consistent with our model that MyoD plays a similar role on MuSC proliferation in all three species.
- Supplementary Fig. 2P: Quantification of SRM protein expression as percent-positive MuSCs. This graph replaces earlier quantifications as MFI.
- Supplementary Fig. 2Q: Quantification of SAT1 protein expression as percent-positive MuSCs. This graph replaces earlier quantifications as MFI.
- Supplementary Fig. 2R: Quantification of OAZ1 protein expression as percent-positive MuSCs. This graph replaces earlier quantifications as MFI.
- Supplementary Fig. 3B: We have added higher magnification, contrast enhanced images to better illustrate the luciferase positive cells.
- Supplementary Fig. 3L: Analysis of CFD protein levels in mouse lemur FAPs following treatment with siRNA against CFD or control. These data confirm our model that mouse lemur FAPs express high levels of CFD. Moreover, this experiment demonstrates that siRNA-mediated knockdown is feasible in mouse lemur cells and that the antibody is in fact specific to mouse lemur CFD.
- Supplementary Fig. 3M: Analysis of adipogenic differentiation of mouse lemur FAPs following treatment with siRNA against CFD or control. These data confirm our inhibitor experiments and show that high levels of CFD drive adipogenic differentiation in mouse lemur FAPs.
- Supplementary Fig. 4H: Analysis of GLIS3 protein staining in MuSCs in cryosection of four species (mouse, mouse lemur, macaque, and human muscles). These images demonstrate distinct GLIS3 protein staining in PAX7-positive nuclei in all species, showing that the antibody detects all species.

- Supplementary Fig. 4I: Quantification of GLIS3 protein staining in MuSCs of four species. These data show high GLIS3 expression in human and mouse lemur MuSCs, and rare expression in mouse and macaque MuSCs, supporting our model that the transcriptomic analyses can predict protein expression levels.
- We provide split channels for the following figure panels: Fig. 1G, H, K; Fig. 2C, E; Fig. 3D, E, F, G; Fig. 4F; Supplementary Fig. 1F, G; Supplementary Fig. 2; Supplementary Fig. 2C, P, Q, R; Supplementary Fig. 3E, F, G, H; Supplementary Fig. 4C, K.

POINT-BY-POINT RESPONSES TO REFEREES' COMMENTS

REVIEWER #1

“This reviewer appreciates the extensive work made on a new animal model, with the purpose to show that it can represent a model closer to humans and therefore more reliable than the mouse model; however, the conclusion that non-human primates represent a model closer to human than mice was highly predictable. This reviewer also acknowledges that the analysis presented have some potential to reveal differences in skeletal muscle properties across animals; yet, they remains underdeveloped at this stage and therefore are not very informative.”

We thank the reviewer for acknowledging the extensive work we put in towards establishing a new animal model. Indeed, it is not surprising that a non-human primate (NHP) more closely resembles humans than mice do. However, we would argue that it was never our intention to surprise. The main purpose for starting this work is the lack of a good NHP model for regenerative medicine that could fill the gap in between mouse and human. Currently, there is a dearth of knowledge on stem cells in NHP, yet the FDA often requires NHP studies as a stepping stone towards clinical trials. Our study will serve as a foundation for future studies of NHP stem cells. The differences in cellular phenotypes mainly serve to illustrate the need for an animal model that more closely approximates human biology than mouse does. Finally, although it is not surprising that a NHP is a better model for human biology than mouse is, it was not clear what aspects or properties of human biology are better modeled in NHP than in mice. We find that the mouse lemur stem cells better approximate the kinetics of human stem cell behaviors than mouse stem cells, as well as metabolic signaling pathways, whereas many other important pathways appear conserved across all three species. This was surprising to us.

- 1) *“There are intrinsic technical limitations that reduce the interest for this work. For instance, the advantage of using this model is limited by the fact that studies are restricted to quiescent cells from post-mortem animals.”*

We respectfully disagree. While our study design led us to use only cells obtained post-mortem, largely because of the need to first develop the specific tools for identification and study of mouse lemur stem cells (e.g. cell dissociation protocols,

cross-reacting antibodies, cell surface markers panels, cell culture conditions), now that we have defined the stem cell populations, future studies can be designed that involve live mouse lemurs. For example, stem cell transplantation experiments can be designed to monitor stem cell engraftment over time in mouse lemurs. This would be an important step towards developing mouse lemur as a NHP model for stem cell transplantation. In addition, studies can be designed in which mouse lemurs are treated with drugs that target the MuSCs in mice, as a type of preclinical study to assess efficacy in the context of primate physiology. Since the size of a mouse lemur is similar to the size of a mouse, such experiments could use similar dosing strategies and would thus be much more affordable than similar studies in the much larger NHPs such as macaques. See for example the following two preclinical studies in mouse lemur: (Rahman A, PLOS ONE, 2015) and (Pifferi A, J Lipid Res, 2015).

Moreover, we would also point out that our study is part of a larger collection of papers on mouse lemurs (including Casey KM et al, Comp Med, 2021; Liu S et al, Nat Comm, 2024; Guethlein LA, BioRxiv; Tabula Microcebus, BioRxiv). Whereas our study was designed to focus on the value of the mouse lemur as a model for stem cell biology, companion papers provide even broader support for the mouse lemur as a NHP model. The mouse lemur consortium (James Briscoe's lab) is working to develop mouse lemur iPSCs.

- 2) *“scRNAseq samples from the two animals used are not homogeneous in term of numbers of single cell isolated and analysed.”*

The difference in numbers of cells sequenced is informed by the study design. Initially, not knowing the success of the approach, we used only a single chamber on a 10x chip, giving us a potential of 4,000 cells sequenced. The second sample was sequenced five years later and this time we opted to increase the number of cells sequenced by loading three chambers, with the aim to uncover low abundant cell types. Importantly, all cell types identified in the second sample were also present in the first sample and at similar relative abundance (i.e. we did not identify any new cell types by sequencing 3x the number of cells). As such, we think that the different numbers of cells sequenced do not impact our findings in any way. For the statistical comparisons of the transcriptomic data, we randomly selected a subset of cells while excluding cell types with fewer than 20 cells sequenced, thus ensuring proper comparisons.

- 3) *“Data comparison with publicly available single cell RNAseq data generated from skeletal muscles from two young (4-year old) crab-eating macaques are problematic in term of age differences with other models”*

We agree with this caveat. It is a challenge also with the human samples, which, although aged, are certainly not from geriatric donors like the mouse lemur and mouse samples. This is simply a limitation of donor availability. Interestingly, given the abundance of wild mouse lemurs, we anticipate that acquiring mouse lemur donors of different ages will become more straightforward than for other NHPs, like

the macaque, which are all endangered and thus highly regulated and highly scrutinized by the general public. We have added a section in the discussion to highlight the challenges of interpreting comparisons of donors with different ages.

- 4) *“The conclusions on comparative analysis are purely speculative, although of potential interest, without a functional follow-up. Perhaps, the unicity of macaque specific cell populations could turn into an interesting point to develop, as it might reveal specie-specific muscle properties to exploit for future therapeutic purposes. However, this requires experimental evidence that are not presented in this manuscript.”*

We agree that this part of the manuscript, the identification of macaque-specific cell types in muscle, is *in silico* only and therefore descriptive by definition and for hypothesis development rather than testing. To further explore these molecular cell types, we tested 2 additional antibody clones against endothelial cell markers (CD31 and CD48, see Supplementary Table 1). Unfortunately, neither of these clones cross-reacted with mouse lemur or macaque muscle (despite proclaimed cross-reactivity), putting direct validation of the endothelial cell subsets by FACS or histology out of reach until antibodies against endothelial cell markers are developed that have good reactivity for the primate species in FACS or histology applications.

Instead, we next looked at other published datasets to find molecular signatures of the macaque-specific cell types. First, whereas the macaque-specific populations of capillary cells and pericytes are present in the macaque skeletal muscle 10x data, such populations do not exist in lung 10x data from the same animals (see Figure 1a,b below). This suggests that the macaque-specific cell types are a unique adaptation of skeletal muscle.

Figure 1: UMAPs of mouse lemur lung single cell RNA sequencing. Cell types were colored by species (a) or by cell type (b), with the endothelial compartment top left.

Second, the macaque-specific cell types are present in published single nucleus RNA seq data of 6-year-old macaques (see Figure 2a,b below). Crab-eating macaques are fully grown young adult animals by the age of 4, making it unlikely that the cell types in this dataset reflect a post-natal growth state of the muscle vasculature.

Figure 2: A,B) UMAPs of macaque muscle single nucleus RNA sequencing. Cell types were clustered by cell type (a). Expression levels of RAMP3, the marker of macaque-specific capillary cells, are plotted in (b). The endothelial compartment is marked by a circle.

Third, we could not find a signature of the macaque-specific cell types in our *Tabula Muris Senis* data (including 1- and 3-month-old mice, see Figure 3a,b below) and the *Tabula Sapiens* data (including a 38-year-old donor).

Figure 3: A,B) UMAPs of mouse muscle single cell RNA sequencing. Cell types were clustered by cell type (a). Expression levels of RAMP3, the marker of macaque-specific capillary cells, are plotted in (b). The endothelial compartment is marked by a circle.

[Paragraph Redacted]

[Figure redacted]

Therefore, although we cannot be certain at this point, we believe that the macaque populations could be a biological feature of macaque muscle independent of age. We have included new sentences in the discussion on these additional in silico analyses in the manuscript. Since these analyses reflect published data, we opted against adding an extra figure to the manuscript.

Finally, to test whether the transcriptomic differences observed between macaque and the other two primates could result in differences in protein expression, we performed immunohistological stainings for GLIS3, a transcription factor we find to be expressed at the RNA level in human and mouse lemur MuSCs, yet absent from macaque and mouse MuSCs (Supplementary Figure 4d). We indeed find nuclear GLIS3 staining in a large fraction of PAX7-positive MuSCs in human and mouse lemur muscle sections, but only in rare MuSCs in macaque and mouse muscle sections (Supplementary Figure 4h,i). These data corroborate the molecular results. While this is of course a different cell type, it supports the idea that a strong differences in expression of an RNA transcript can result in the presence of absence of the corresponding protein in the histology.

- 5) *There are many genes that appear not expressed/detected in some specie, despite their function has been well established and essential for muscle development and homeostasis. It would be interesting to determine whether functional paralogs are expressed that mediate similar functions across species.”*

We thank the reviewer for this suggestion. This is a clear application of our in-silico datasets and another important demonstration of how the field could make use of a larger variety in animal models. For example, primate myofibers express higher levels of MYBPC1 and MYL3, whereas mouse myofibers express higher levels of the respective paralogs MYBPC3 and MYL4. We are also intrigued by the instances

where a gene seems to switch between cell types (i.e. it is expressed in one cell type in one species, yet in a different cell type in another species), which would suggest a dynamic regulation among species that is currently underappreciated. For example, *CACNA1S* is expressed in myofibers in all species but also in T cells in mice. Interestingly, examples of this phenomenon have also come out of other atlas studies. For example, the human lung cell atlas revealed that two key risk genes for COPD/emphysema (*SERPINA1* and *HHIP*) are selectively expressed in AT2 cells in human but in alveolar stromal cells in mice (Travaglini et al Nature 2020). Moreover, two companion papers have explored these points at a more global level. In those papers, we report that the vast majority of genes (89%) displayed evolutionarily divergent expression patterns between the four tested species (Tabula Microcebus Consortium, BioRxiv). Expression conservation did not correlate with coding sequence conservation, implying separate evolutionary diversification mechanisms and/or selective pressures for the expression control sequences and protein coding sequences at each gene. For example, the potassium channel *KCNK3*, mutations in which cause pulmonary arterial hypertension in humans, was selectively expressed in lung pericytes (but not muscle pericytes) of human, mouse lemur, and mouse, but not of macaque. We have added a section to the Discussion on paralogues, which also references the companion papers.

- 6) *“The authors mention that some antibodies work because of their binding to conserved epitopes, but nowhere in the manuscript they provide protein alignments for the three different species”*

We have now included multiple sequence alignments of the antibody epitopes in Supplementary Table 1.

- 7) *“The conclusion that overexpression of MYOD in lemur satellite cells leads to activation of the cells is based on the increased expression of MYOD in MYOD transfected cells compared to GFP control. This is not surprising and should be corroborated by analysis of downstream transcriptional targets of MYOD”*

This is a challenging experiment to do. It remains an open question whether MyoD has the same target genes in activating mouse lemur MuSCs as it has in activating mouse MuSCs. On this note, we would also point out that all of the canonical target genes of MyoD have been described in the context of *differentiating* (e.g. *MYOG*) or *proliferating* (e.g. *ID3*) *myoblasts*, and not in activating MuSCs. We make this point only to highlight that a negative result in terms of the expression of known MyoD target genes from mouse myoblasts does not in any way refute the hypothesis that MyoD drives activation of mouse lemur MuSCs, nor does an increase in the expression of these genes prove the hypothesis. However, we overexpressed MyoD in proliferating mouse lemur myoblasts and measured the canonical MyoD target gene *ID3* by quantitative RT-PCR. Indeed, *ID3* is upregulated in response to MyoD overexpression in the context of proliferating mouse lemur myoblasts (Supplementary Figure 2I).

- 8) *“It would be important to show also the % of cells expressing protein involved in the spermidine pathway and not only the total intensity of IF”*

We agree with the Reviewer that marking cells as positive or negative offers a more robust way of quantifying protein expression across different species than using raw fluorescence intensity measurements. We have now included graphs depicting the fraction of positive cells (Supplementary Figures 2p-r). The new graphs replace the original graphs showing mean fluorescence intensity and fully support our previous conclusions.

REVIEWER #2:

“This is an interesting manuscript that delves into characteristics of lemur mouse muscle stem and progenitors as well as mesenchymal (fibro-adipogenic cells). It describes various comparisons between lemur muscle cells to those of the mice and human counterparts and makes interesting conclusions about the suitability and limitations of each model organism and its relevance in the human context.

We thank the Reviewer for highlighting the general interest of our manuscript.

1. *“Low number of Pax7 expressing cells in the lemur scRNA-seq data is not well explained or justified. Most of the analysis regarding scRNA-seq is global clustering. It would be interesting to perform cell-specific clustering to reveal heterogeneity in lemur, mouse and human muscle stem and FAPs cells.”*

The limit of detection for SmartSeq2 (SS2) sequencing is unknown, but it is certainly not at a single molecule level. Previous single molecule RNA FISH experiments from our labs and others showed that a large fraction (20-30%) of mouse MuSCs express only 1 or 2 RNA molecules for PAX7 (de Morree et al PNAS 2017, de Morree et al Science 2019, Kann et al Development 2019) and such cells would likely not reach the limit of detection for either SS2 or 10x RNAseq and thus show up as negative for PAX7 (so-called sequencing drop-outs). Accordingly, in our human (Tabula Sapiens, Science 2022) and mouse (Tabula Muris, Nature 2018, 2020) datasets, we find that only ~65% and ~20%, respectively, of the FACS-purified MuSCs have detectable PAX7 transcript. Our data for mouse lemur muscle (~80% of MuSCs are positive for PAX7) are entirely in accordance with those observations. Moreover, published single cell RNAseq studies by other labs similarly report a substantial fraction of MuSCs to have PAX7 levels below the limit of detection (Di Micheli et al., Skeletal Muscle 2020; Lovric et al., Communications Biology 2022). Finally, it is well established that there is a wide range in PAX7 expression, as elegantly shown in FACS studies (Rocheteau et al Cell 2012). Obviously, one could increase the sensitivity of the RNAseq assay by increasing the number of amplification cycles. We decided against this to avoid the technical

artefacts this might cause. In summary, we argue that the SS2 data reveal a level of purity similar to (or even higher than) what is seen in mouse and human studies. To further corroborate purity of our sorted stem cells, we stained purified MuSCs and FAPs for PAX7 or PDGFRA protein, respectively, and find that more than 80% have high protein levels (Supplementary Figure 1f,g). Moreover, in 3.5 days of low serum culturing, none of the mouse lemur FAPs stained positive for MyoD (Supplementary Figure 1l), a time point at which $\frac{3}{4}$ of the mouse lemur MuSCs stained positive for MyoD. This suggested that the FAPs are not contaminated with MuSCs.

Our data does not lend itself for clustering analyses (like X-shift or k-nearest neighbor) for two reasons. First, we have not identified the genes to perform the clustering with. Second, those algorithms have been developed for protein-based assays such as CyTOF. Instead, we performed dimension reduction analyses with a PCA algorithm and found that MuSCs from all four species nicely group (“cluster”) together in the UMAP graph, suggesting that the variation among cell types is larger than the variation among species for a given cell type. When exploring the MuSCs for each species using this approach, as the Reviewer suggests, we did not observe distinct subclusters (as we already reported for the human and mouse data), nor did we observe subclusters in the integrated UMAP space. Obviously, sequencing more cells could lead to the identification of subclusters with slightly different molecular signatures, although assessment of their biological function would always depend on the existence of cell surface markers that would enable the purification and subsequent study of such subclusters.

2. *“The FACS strategy in the paper is unclear and needs to be better described and demonstrated. From this reviewer’s understanding, as shown in extended Figure 1E, it appears that the lemur muscle stem cells are only identified and isolated with CD56 (NCAM1), which is a known marker used in human MuSC isolation. However, only using a single antibody is not appropriate for isolating a pure population of cells. Several publications isolate human muscle stem cells, using negative markers (CD11b, CD31, CD34, CD45) along with 1 or 2 positive markers (NCAM1, CXCR4, and CD82). This is also seen in the FAPs isolation, the authors appear to be indicating that the FAPs are isolated using only a THY1 antibody, which beyond the issues identified above of only using one antibody for isolating a pure population, THY1 is a well known marker of fibroblasts.”*

We apologize for the confusion. We have added further clarification to our descriptions of the FACS strategy. The MuSCs are identified by CD56 as a positive marker and CD90 as a negative marker. Conversely, the FAPs are identified by CD90 as a positive marker and CD56 as a negative marker. To further test the purity of the cell populations, we stained the MuSCs and FAPs with antibodies against PAX7 and PDGFRA, respectively. We found that >85% of MuSCs expressed high levels of PAX7 protein and that >80% of FAPs expressed high levels of PDGFRA protein (Supplementary Figure 1f,g). Moreover, where the large majority of MuSCs expressed high levels of MyoD 3 days after serum deprivation

(Figure 2b,c), none of the FAPs did, suggesting that few, if any, MuSCs contaminated the FAP population (Supplementary Figure 11).

3. *“Negative markers (CD31, CD45 and SCA1) are used in FACS experiments testing the use of the known positive marker VCAM-1 (extended Figure 1A). However, it does not appear that they were used afterwards. This may explain the poor purity results that the authors obtained from their sorting of lemur MuSCs, from RNA-Seq only 80% expressed markers associated with MuSCs, while when attempting to culture the cells, only 20% were capable of clonal growth. With similar results seen in the FAPs. The authors must repeat these experiments using a proper FACS antibody panel, including negative selection markers and preferably a second positive marker.”*

As commented above (point #1), we do not think the SS2 data demonstrate poor purity. On the contrary, our drop-out rate for PAX7 is remarkably low. The 20% clonal expansion is entirely in line with our observations on aged mouse MuSCs (Brett Nature Metabolism 2020). It is important to note that aging causes a decline in MuSC function and even then, at best 30% of young mouse MuSCs can clonally expand in this assay. We also note that negative selection is essential for purifying mouse MuSCs because the positive marker, VCAM1, is also expressed on the much more abundant endothelial cells. Judging from our single cell RNAseq data (see Figure 5 below), this appears not to be the case for NCAM1 in mouse lemur muscle. In fact, the venous cells expressing NCAM1 also express THY1, and accordingly we excluded double positive cells from our FACS scheme. We have now screened 23 additional antibody clones with the aim of identifying more cross-reacting antibodies for FACS, which was ultimately unsuccessful. These results are added to Supplementary Table 1. Finally, it is worth noting that it is a long-term goal of the mouse lemur consortium to develop mouse lemur-specific antibodies, especially antibodies targeting cell surface proteins.

Figure 5: Dot blot of cell surface marker expression across muscle cell types in human (H), lemur (L), and mouse (M).

4. *“The authors claim that the human specific antibodies typically used in MuSC FACS did not stain the Lemur muscle stem cells, however no explanation was given as to the cause. It would be important to investigate the epitope that these antibodies recognize and determine whether that region is conserved between human and lemurs. Further, the*

authors should investigate whether the antibodies used to sort mouse MuSCs would be appropriate for use in the lemur context.”

We have included multiple sequence alignments of the known epitopes for all antibodies used in our study (Supplementary Table 1). These results show that for all cross-reacting antibodies the epitope is either 100% identical or close to 100% identical. Unfortunately, for the remaining antibodies we tested, as for most antibodies in general, the epitope remains unknown, or at least undisclosed by the company. Our best guess would be that the antibodies that do cross-react, recognize a stable epitope (i.e. an epitope consisting of a conserved amino acid sequence or of a conserved post-translational modification such as glycosylation), whereas the antibodies that do not cross-react, recognize a variable epitope. We have included data on flow cytometry experiments using the common mouse FACS antibodies, revealing that they do not cross-react to mouse lemur cells (Supplementary Figure 1a,b). In addition, we tested 23 additional FACS antibody clones. This includes 2 clones against cell surface markers expressed in endothelial cells in the RNAseq data (CD31 and CD48) and 19 clones against cell surface markers expressed in immune cells in the RNAseq data (CD3E, CD4, CD7, CD8A, CD14, CD19, ITGAM, SCA1, KIT, IL7R, IL3RA, MS4A1). None of these clones showed any reactivity to our mouse lemur single cell suspensions (Supplementary Table 1). This is not surprising, since it is well established that isolation of human or mouse cells often requires species-specific antibody clones (a benefit of the species-specific antibody clones has been that it enables cross-species engraftment assays). Even though some of the tested clones have been reported in NHP studies (information that we have now added to Supplementary Table 1), we argue that mouse lemur-specific antibodies against CD31 and CD45 will be needed for the confident selection of mouse lemur endothelial and immune cells, respectively. On this note, we would add that it is a long-term goal of the mouse lemur consortium to develop such mouse lemur-specific antibodies. We now provide a table with a longer list of antibody clones in Supplementary Table 1, including information on the reported species reactivity for each clone, and add a section to the Discussion on the development of FACS panels.

5. *“Without being able to clearly demonstrate the isolation of a pure population of cells, subsequent experiments may be in question as any differences observed could simply be attributed to the fact that the cells isolated from the lemur are not pure MuSCs nor FAPs. In order to determine whether the cells sorted were functional muscle stem cells, the authors used lentiviral infection to have the cells express luciferase and GFP. Afterwards the cells were transplanted into a host mouse, then retransplanted into another host mouse. While the authors see an increase in BLI intensity after transplantation and that persists after retransplantation, these results are not conclusive. All they indicate is that cells derived from lemur muscle can populate and persist in mouse muscle. There is no evidence that these are muscle stem cells. It would be essential for the authors to take cross sections of the transplanted muscle and stain for Pax7 and Laminin, to determine whether the transplanted cells are able to home to the muscle stem cell niche and populate the correct anatomical location. The presence of GFP+/PAX7+ cells under the laminin of a myofiber would be indicative of muscle stem cells. The*

presence of luciferase positive myofibers only indicate that some of the cells were at the very least myogenic progenitors, but it is not conclusive that they were muscle stem cells. As it stands, the results from the BLI experiment are open to interpretation.

As the Reviewer suggested, we have analyzed muscle cryosections for GFP+ cells in the satellite cell position, and we now include an example (Supplementary Figure 1q). However, we would also note that serial transplantation is the gold standard for stem cell function. We are unaware of any data in the MuSC literature in which serial transplantation was successful without the initial transplantation resulting in the establishment of quiescent stem cells in the muscle stem cell niche. In further support that this is the case for our studies are the data showing that at least some of the transplanted cells retain CD56 expression, the marker of quiescent MuSCs (Supplementary Figure 1o).

6. *“The luciferase staining of the transplanted muscle cross section is not convincing (extended Figure 1N). The signal may simply be background, it is necessary to show at least one negative myofiber as it defies expectation that every single myofiber regenerated using the transplanted cells, and to the same degree to have the same level of luciferase, considering there was only 1000 cells transplanted. The likelihood that these luciferase positive myofibers are simply due to background is supported by the cross section staining in extended Figure 3L, where luciferase expressing FAPs were transplanted into the muscle. Yet we see that every myofiber is positive for luciferase, despite the fact that FAPs do not fuse with myofibers and therefore the myofibers cannot have obtained luciferase expression.”*

We now extended the field of view revealed in the image to include fibers more distal to the injury site that are negative for luciferase (Supplementary Figure 1p, formerly extended Figure 1N). We also performed a background correction on all images to scale back the autofluorescence signal, which we had not performed on the image of the FAP transplantation (Supplementary Figure 3b).

7. *“The growth media used in this paper for MuSC/myoblast growth is F10 supplemented with 20% HS, 1% P/S and 2.5 ng/mL of FGF. To this reviewer’s knowledge there is no published paper using this specific media for the growth of MuSCs or myoblasts in culture. Did the authors mean 20% FBS instead, and if not what is the justification for using 20% HS.”*

We indeed used 20% HS as we described previously (Brett et al Nature Metabolism 2022). We agree that the common growth medium uses FBS. However, we observed that when culturing small numbers of cells, such as in the single cell myoclone assay, the 20% HS leads to better survival. This would suggest that the cells have an optimum concentration of growth factors, which depends in part on the environment. Such a concentration optimum has been well-established for FGF in the culturing of mouse myoblasts (Rando and Blau JCB 1994). However, to grow

the myoblasts, we use the common medium DMEM/F10 (1;1) + 20%FBS + 1% P/S + FGF, which we had not included in the methods. We have now added this information.

8. *“Please show the individual channels for all staining pictures. Particularly the MyoD staining for cultured cells and the TA cross sections.”*

We have now added single channels for all stainings throughout the manuscript.

9. *“Can the authors clarify why they performed a 14 day differentiation assay of the myoblasts when in humans 6-8 days is the typical time to fully differentiate and in mice 5-7 days is sufficient.”*

The differentiation timeline was inferred from cell culture observations. The cells did not become contractile until two weeks in differentiation medium. This may be due to the species, but the age of the cells might also limit their differentiation. In contrast, the standard differentiation times the Reviewer refers to relate to the expression of MHC in multinucleated myotubes. We opted for an extended time, first, because we sought the functional assessment, and second, because we did not want to be fully dependent on a protein stain for determining the maturation of the myotubes.

10. *“In extended Figure 3L, the legend states that these are TA cross sections of NSG mice that had MuSCs transplanted in them. However, in the text it says that this experiment was done using transplanted FAPs. Please correct this.”*

We thank the Reviewer for catching this error. We have now corrected the text of the figure legend (now Figure 3i).

11. *“The authors write “To test whether MyoD can drive mouse lemur MuSC activation, similar to its role in mouse MuSCs, we overexpressed recombinant mouse MyoD in freshly isolated mouse lemur MuSCs. This caused increased MyoD protein expression and increased MuSC activation after 24h (Extended Data Fig. 2f,g).” This statement does not reflect the experiment performed as EDU was added for 36 hours, not 24 hours, and the experimental design is also unclear. From this reviewer’s understanding, muscle stem cells were isolated and then transfected with either MyoD or GFP using lipofectamine, transfection itself takes between 1-3 days to be able to see overexpression of the protein of interest. Therefore, I do not understand whether the EDU was added during the transfection step, thereby causing the EDU and the transfection to occur simultaneously, which would not be ideal as the cells would not be over expressing MyoD initially and would be growing in transfection reagent. Or was the EDU added after transfection was completed, in which case this experiment cannot be assessing the role of MyoD in activation as the cells would have already activated by the time the EDU was added. Further, the increase in EDU positive cells does not appear to be biologically significant as the control condition sees approximately 0.6% positive cells, while the MyoD O/E has about 1.2% positive. This is an increase that does not come near the*

levels seen in the WT mouse MuSCs at 36 hours (~25% EDU positive), therefore the claim that mouse MuSCs activate faster due to their increase MyoD levels is not supported by this experiment. Lastly, there are no representative staining pictures which are necessary to validate the claims made by the authors.”

We apologize for the confusion and have now corrected the experimental description (now **Supplementary Fig. 2j,k**, formerly Extended Data Fig. 2f,g). Cells were transfected overnight, and after 12 hours the medium was refreshed and EdU added. The cells were then cultured for an additional 24 hours. Accordingly, the cells received a 24h EdU pulse, yet the time of measurement was 36h following isolation and 24h following transfection.

We agree with the Reviewer that the effect is mild. This is likely due to limited transfection efficiency. We have now quantified the number of GFP positive cells and find a transfection efficiency of ~10% (see Figure 6 below). The clear detection of GFP demonstrates that transfection on this time scale is in fact feasible. Moreover, this is entirely consistent with our previous work on mouse MuSCs, where we could observe efficient expression of recombinant MyoD in 24h, resulting in a similar doubling of the fraction of EdU positive cells at that time point (de Morree PNAS 2017).

We opted against including images of the transfected cells, due to the low confluency at which the experiment was run and the low magnification at which it was imaged. Practically, this means having a single cell per field of view and we do not think showing a single cell would be informative. We did include an example of this staining in the Figure 6 below.

Figure 6: Mouse lemur MuSCs were transfected with GFP, cultured in the presence of EdU, and stained for GFP (arrow) and EdU (arrowhead). Example images and quantification to the right.

12. *“The materials and methods sections are missing some important details. For examples, on page 15, line 563 the authors state “Single cells were seeded in 96-well plates coated with ECM (1:100)” which ECM? Catalogue number?”*

We have added the missing catalog number (Engelbreth-Holm-Swarm murine sarcoma from Sigma, E1270). In addition, we carefully read through the methods and added additional catalog information.

13. *“Page 15, line 582, the author state “For fibrogenic differentiation, cells were 582 maintained in fibrogenic medium (10% FBS, 1% Pen/Strep, 1 ng/ml TGFβ1)” what’s the base medium? Is it DMEM?, F10? Something else?”*

We have added the base medium (F10).

REVIEWER #3:

“In general, the research is engaging. The majority of findings are well-organized and trustworthy, aligning with the standards of “Nature Communications”. Nevertheless, there is potential for further enhancement if feasible. More information can be found below:”

We thank the Reviewer for the compliments.

1. *“Lines 80-81 describe that the authors analyzed 3,122 single cells from one sample and 9,409 single cells from another sample in their single-cell sequencing (Fig. 1b). How to explain the significant difference between these two samples?”*

See also our response to point #2 of Reviewer #1. The samples come from the first and the final lemur, which were sequenced with 5 years in between. For the first sample, we used only a single chamber on the 10x chip. For the second sample, we opted to increase our yield and used 3 chambers. Importantly, there is no difference between the two samples in terms of cell types identified and relative abundance of cell types.

2. *“Line 86-88, the authors described antibodies used for purification of mouse MuSCs were not effective for labeling lemur MuSCs (Extended Data Fig. 1a, b). Consider using humanized antibodies to retry this assay.”*

We have now included the results from flow cytometry experiments using 23 additional antibodies against human target proteins, none of which had reasonable staining against the mouse lemur cell suspensions (Supplementary Table 1). We are unfamiliar with the use of humanized antibodies in FACS. Such antibodies would presumably combine a human Fc domain with the VH domain of another species

such as mouse. Such antibodies would be valuable for therapeutic use but offer no apparent benefit over the conventional antibodies in flow cytometry experiments. We searched for humanized antibodies in the BD catalog and could not find any humanized antibodies available for use in FACS.

3. *“Line 170-176, the authors concluded that NCAM1+/THY- cells are true MuSCs that can engraft, differentiate, and self-renew (Fig.1l). How about the function of newly formed myofiber? Please make it clear.”*

It would be difficult, if not impossible, to assess the function of donor-derived myofibers in vivo, simply because the majority of the surrounding fibers is host-derived. Instead, we have now added a video of contracting mouse lemur myotubes to demonstrate the function of the differentiated MuSCs ex vivo (Supplementary File 1). Moreover, we include immunostainings to show that the myoclone-derived myotubes and the expanded MuSC-derived myotubes are mature myofibers that are MYH2-positive (Figure 1j and Supplementary Figure 1j).

4. *“Line 190-193, the authors concluded fewer mouse lemur or human MuSCs exhibited MyoD protein expression during MuSC activation (Fig.2b). The results of immunostaining alone are not sufficient to support this conclusion. Please include western blot or mass spectrometry experiments to strengthen these findings.”*

We agree with the reviewer that western blot is the gold standard for protein detection. However, we would also argue that quantifying MuSCs as MyoD-positive or negative is the established approach for assessing MyoD expression and/or MuSC activation. Unfortunately, even if we had an antibody that worked in western blot on all three species, it is near impossible to detect a protein by western blot that is expressed by at best 5% of the stem cells in mice, let alone to detect it in mouse lemur cells. In fact, in our previous work, we were unable to detect MyoD protein in freshly isolated mouse MuSCs using conventional or capillary westerns (de Morree PNAS 2017), even though up to 5% of MuSCs were positive by IF and by immunohistochemistry, which is entirely in line with our current IF result for mouse MuSCs (Supplementary Figure 2g). Accordingly, the quantification of the number of MyoD-positive cells by microscopy has been the de facto standard. Moreover, we would note that mass spectrometry or western blots on quiescent MuSCs require cells from multiple mice as input (Benjamin et al Cell Metabolism 2023; Salvi et al PNAS 2022). Unfortunately, the availability of mouse lemurs is limited. If we had more animals and were using them for other experiments, we could certainly plan to do western blots as well.

Nevertheless, to further support our data on MyoD expression, we performed single cell westerns on freshly isolated mouse MuSCs, confirming that 2-4% of the MuSCs are positive for MyoD protein (Supplementary Figure 2h), corroborating the immunofluorescence analysis. In addition, we analyzed our published CyTOF data (Porpiglia et al Nat. Cell. Bio. 2017) of single cell suspensions from uninjured mouse muscles (Supplementary Figure 2i), confirming that 2-5% of the MuSCs are

positive for MyoD protein, again corroborating the immunofluorescence analysis. Finally, we included multiple sequence alignments of the antibody epitope showing that the epitope is identical in all four species (Supplementary Table 1).

5. *“Line 246-253, the authors indicated that lower spermidine levels in primate contributed to delayed activation phenotypes of MuSCs (Fig.2j). Would adding spermidine to MuSCs in primates restore the delayed phenotype?”*

We have added spermidine to freshly isolated human MuSCs and observed a more rapid rise in the number of EdU-positive cells compared to control (Figure 2m). This is consistent with our model that spermidine levels regulate MuSC quiescence and activation.

6. *“Line 213-214, the authors described that primate MuSCs showed higher levels of RNA expression of NOTCH target genes HES1 and HEY1 (Extended Data Fig. 2c). From the graph, we can see that HES1 expression is extremely high in the mouse lemur, while HEY1 expression is comparable across all three species. Please explain it.”*

In Supplementary Figure 2e (formerly Extended Data Figure 2c), we show that human and lemur MuSCs have higher expression of HES1 and HEY1 compared to mouse MuSCs. However, HES1 expression was particularly high in mouse lemur MuSCs. We would imagine that the expression differences reflect species-specific regulation. Since these genes can act as cofactors of other transcription factors, this could result in differences in downstream signaling. For example, HES1 can act as a co-factor of MyoD in the regulation of Pax3 gene expression in mouse iPSC-derived myogenic cells (Sato et al Stem Cell Reports, 2019). However, as we note elsewhere in the rebuttal letter and in the manuscript, we prefer to focus on cases where a gene is either present or absent because in those cases it is more straightforward to speculate on implications for cell function.

7. *“Line 216-217, the authors concluded that Primate MuSCs expressed lower levels of MyoD1 RNA compared to mouse MuSCs. Please confirm this result by quantitative real-time PCR.”*

We have now included microfluidic PCRs to confirm the low levels of MyoD expression in mouse lemur MuSCs compared to mouse MuSCs (Supplementary Figure 2f).

8. *“Line 235-239, the authors indicated that protein levels of OAZ1, SAT1 and SRM are higher in primate MuSCs compared with mouse (Extended Data Fig. 2k-m). There is an inconsistency between the microscopic images and the statistical data. Please include western blot or mass spectrometry experiments to enhance the reliability of these results.”*

We have included new images to better reflect the mean quantification (Supplementary Figures 2p-r). It is near impossible to perform western blots on

quiescent MuSCs, which in the case of mouse MuSCs require cells from multiple animals as input. Moreover, the availability of mouse lemurs is limited, nor do we have antibodies for any of the target proteins that work in western blot. Instead, to confirm the expression differences in OAZ1, SAT1, and SRM that we uncovered by RNA sequencing and IF staining, we stained for the downstream metabolite spermidine. As predicted, primate MuSCs, which have lower levels of SRM, the rate-limiting enzyme for spermidine biosynthesis (Supplementary Figure 2p), and which have higher levels of SAT1, the rate-limiting enzyme for spermidine breakdown (Supplementary Figure 2q), compared to mouse MuSCs, also have lower steady-state levels of spermidine (Figure 2d). Moreover, we confirm the expression of SAT1 by treating MuSCs with a small molecule potentiator, which indeed resulted in lower spermidine levels (Figure 2f-h).

9. *“In Fig.3d-3g, the immunostaining results do not sufficiently demonstrate the differences in adipocyte markers among the three species due to notable discrepancies between the microscopic images and the statistical data. Please incorporate quantitative experiments, such as mass spectrometry, to clarify these differences.”*

We have included new images to better reflect the mean quantification (Figure 3d-g). In addition, we confirmed the gene expression differences in Perilipin1, SMA, and RUNX2 with histochemical stains for adipose deposits (Oil Red O, Supplementary Figure 3e), fibrotic deposits (WGA, Supplementary Figure 3f), and calcium deposits (Alizarin Red, Supplementary Figure 3g), respectively. In addition, we confirmed the expression of CFD protein with a small molecule inhibitor, which we found to be effective in limiting adipogenesis in mouse lemur FAPs, which express CFD protein (Supplementary Figure 3j; Figure 3g), but not in mouse FAPs, which do not express CFD protein (Supplementary Figure 3k; Figure 3g). Finally, we treated mouse lemur FAPs with siRNA against CFD (see also our response to point#10 below), which resulted in a significant reduction in staining intensity, demonstrating the specificity of the antibody (Supplementary Figure 3l). As mentioned above, performing quantitative mass spectrometry requires protein extracts pooled from MuSCs from multiple animals. Due to the limited availability of mouse lemurs, and the lack of antibodies working in western blotting, we were unable to perform western blotting.

10. *“Line 302-315, the authors evaluated CFD's role in FAP adipogenesis using a small molecule inhibitor to block CFD activity. Due to potential negative effects of the inhibitor on cells, generating CFD knockout or knockdown cell lines would provide a clearer understanding of CFD's role both in vitro and in vivo.”*

We performed knockdown experiments to confirm the role of CFD in adipogenic differentiation. We serially transfected mouse lemur FAPs with siRNA against CFD or control (day 0 and day 3). We replaced the transfection medium with adipogenic medium 6 hours following transfection. We fixed one plate of cells on day 4 to measure CFD protein levels by immunofluorescence and observed a significant decline in protein staining intensity confirming both the knockdown and antibody

specificity (**Supplementary Figure 3l**). A second plate was kept in adipogenic medium, with the medium being refreshed every three days until day 14 when the cells were fixed and stained for Oil Red O. The siRNA against CFD caused a decline in Oil Red O staining, confirming the role of CFD in driving adipogenesis (**Supplementary Figure 3m**).

11. *“Line 323-330 conclude that there are two unique cell populations present in macaques (Fig. 4a-b). However, the single-cell sequencing data for macaques come from young individuals, while data for mice, humans, and lemurs are from older individuals. Could the presence of these two cell populations in macaques be attributable to age differences? Please clarify this point.”*

We agree with the Reviewer that the two molecular cell types we identify in macaques could be attributed to these samples coming from young animals. However, we don’t think this is the case (see also our response to point #4 of Reviewer #1). The populations are not found in lung tissue from the same macaques. Moreover, they do appear in the single nucleus RNAseq data from 6-year-old macaques, whereas they do not appear in tabula muris (1- and 3-month-old mice included) and tabula sapiens (the youngest donor sequenced is 38 years old) datasets. Nevertheless, we cannot exclude the possibility that the molecular cell types do reflect a state unique to young adult muscle and we have added a section in the Discussion to highlight this.

12. *“Line 402-405 describe higher levels of SGCA expression in purified primate MuSCs at the protein level (Extended Data Fig. 4i). However, the immunostaining images for mice and lemurs do not show noticeable green fluorescence. Please include western blot or mass spectrometry experiments to validate this conclusion.”*

We have performed deconvolution to bring out the signal of the SGCA (Supplementary Figure 4k**). It is near impossible to perform western blots or mass spectrometry on quiescent MuSCs, which require cells from multiple animals as input. If we had more mouse lemurs available, and we had an antibody that worked in western blot, we could certainly plan for western blot experiments. Unfortunately, neither is the case.**

[Paragraph redacted]

[Figure redacted]

13. *“Line 413-420, the authors described FAM111B is expressed at low levels in primate capillary endothelial cells and MYL5 is expressed in primate MuSCs and myofibers (Extended Data Fig. 4j). However, the expression of FAM111B and MYL5 was not observed in the primate cells shown in the bubble graph. Please clarify this discrepancy.”*

We agree with the Reviewer that the expression is barely visible in the pdf rendering. We have exchanged the figure for a table (Supplementary Table 8) with the gene expression levels.

POINT-BY-POINT RESPONSES TO REFEREES' COMMENTS

REVIEWER #1

“The authors have fairly addressed most of the reviewer comments and the revised version of the manuscript is definitely improved”

We thank the reviewer for acknowledging how the manuscript has improved.

REVIEWER #2:

“The revised manuscript shows many improvements, however, there are still some issues of clarity and the need for further refinement.”

We thank the reviewer for acknowledging the many manuscript improvements.

- 1) *“The one major issue remaining is the isolation of the cells of interest (MuSCs and FAPs) from the lemurs. This reviewer is still not convinced on the purity of the cells isolated, mainly due to the FACS strategy. Doing a single positive selection with no negative selection raises doubts on the identity of the sorted cells. While I could be convinced on the NCAM1 sorting for the MuSCs as it is not a common marker, mostly seen in neurons and it is highly expressed in MuSCs and myogenic cells. However, that is not the case for THY1. While it is expressed in FAPs, it is also expressed in a multitude of other cell types, many of which are present in the muscle, including fibroblasts, adipocytes. The confirmatory staining of Pdgfra is not particularly useful in this case as fibroblasts can express Pdgfra as well. The purity of the lemur MuSC and FAPs isolation should also be compared to the purity the authors see from both the mouse and human samples. Until the authors can fully validate the cells they are using, no solid conclusion can be derived from all subsequent experiments.”*

PDGFR α is the gold standard marker for FAPs in mice. We had included purity data showing that 1) >75% of the sorted THY1^{+ve}/NCAM1^{-ve} were positive for PDGFR α mRNA in smartseq2 single cell RNAseq; 2) the sorted THY1^{+ve}/NCAM1^{-ve} cells were positive for PDGFR α protein in immunofluorescence staining; and 3) none of the cell cultures derived from sorted THY1^{+ve}/NCAM1^{-ve} yielded any MyoD positive cells (a defining marker for myogenic cells, the most likely contaminant cell type). Per the Reviewer's suggestion, we have now included purity data based on immunofluorescence stains of mouse MuSCs (90% PAX7^{+ve}), human MuSCs (85% PAX7^{+ve}), mouse FAPs (95% PDGFR α ^{+ve}), and human FAPs (90% PDGFR α ^{+ve}). These numbers are comparable to what we observed for the purified mouse lemur MuSCs (85% PAX7^{+ve}) and mouse lemur FAPs (80% PDGFR α ^{+ve}). In addition, we have amended the text to clarify that the THY1^{+ve}/NCAM1^{-ve} cell population is enriched for FAPs.

We would never argue that any cell population isolated by FACS is 100% pure. Even in the case of the highly established protocols for the isolation of mouse

MuSCs, we find cells that do not express PAX7 mRNA and protein (Liu et al Nat Prot 2015; de Morree et al PNAS 2017). While this could in part be explained by detection limits and biological variation (for example, a subset of MuSCs on fiber explants are negative for PAX7 mRNA as detected by single molecule FISH (Kann et al Development 2019), we would certainly add that antibody-based cell sorting is not flawless and that non-MuSCs are likely to be included among the sorted cells. We would however also argue that the validity of our studies *in no way* requires an absolute purity of the cell populations. All of our data are based on population averages, not single cells. Since we observe comparable purity scores for the three species, we argue that the population averages can be directly compared.

To further support the specificity of PDGFR α as a marker of FAPs in mouse lemur, we plotted PDGFR α expression across all cell types for the three species. As the Reviewer points out, and our data confirm, in mice, PDGFR α mRNA marks several cell types in addition to the FAPs, such as pericytes and tenocytes. In contrast, in our mouse lemur single cell RNAseq data we only detect PDGFR α mRNA in the FAPs. This strongly suggests that PDGFR α is a good marker for mouse lemur FAPs.

- 2) *“The TA PAX7 staining looks fine, but the claim in the manuscript is that these cells are in a sublamina position “Furthermore, we observed PAX7+ cells in the MuSC-position under the basal lamina in mouse lemur muscle cryosections”, yet no laminin staining was done, therefore the authors cannot claim that these are PAX7 positive cells in the proper anatomical location. To fully confirm that this PAX7 staining is labeling MuSCs there must be a laminin staining as well. There is a consistent issue throughout the paper, anytime a cross-section staining for PAX7 or GFP (in the case of the BLI transplantation experiment), the cross-section has only WGA counterstaining to mark cell membranes and the WGA staining is not particularly clean either. However, that is not sufficient to show that these are MuSCs in the proper location as the positive cells could easily be in the interstitial space. The transplantation staining of GFP cells should ideally be redone with laminin and PAX7 to properly show whether these GFP cells are in the correct anatomical location and are expressing the MuSC marker PAX7.”*

We apologize for this mistake. We have amended the text to omit the reference to the basil lamina. We fully agree with the Reviewer that WGA is not a basal lamina stain. WGA stains the collagen matrix surrounding the basal lamina and the outer membrane. It is therefore not surprising that the staining pattern of WGA is less distinct than one would expect from a laminin stain. The main reason for using WGA instead of a laminin stain is the lack of an antibody that cross-reacts with

mouse lemur laminin following the antigen retrieval process that is required to enable the PAX7 protein staining. This may partly be due to technical challenges (e.g. epitope conservation or availability). However, it may also reflect a biological feature of mouse lemur muscle. Judging from our gene expression data, the RNA levels of genes encoding matrix proteins are variable across the three species. Laminin is expressed at much lower levels in the mouse lemur myofibers compared to mouse and even human myofibers. See for example the figure below for the laminin-2. This may result in lower levels of deposited laminin in mouse lemur basal lamina, which in turn could explain the challenges with staining for mouse lemur laminin protein. This is of course entirely speculative, and therefore we decided against discussing this in our manuscript. Interestingly, our data also suggests that laminin is produced by different cell types in the three species. This could indicate differences in biosynthesis of the basal lamina between species, a process that remains poorly understood.

Among our microscopy images we do find examples in which the myofibers have separated and where a MuSC remains associated with the separated fiber. This suggests that at least some of the cells exist in a sublaminar position. We have included an example below. Finally, we would point out that we do not know of a single example in the MuSC literature where PAX7⁺ cells were interstitial under homeostatic conditions.

- 3) “After FACS sorting the MuSCs, they are stained for Pax7, and FAPs were stained for Pdgfra. While these cells were cytopun onto a slide, it is not stated in the manuscript, only in the methods. It would be clearer to say that freshly isolated MuSCs and FAPs were stained. Further, no quantification is given for the percent Pax7⁺ or Pdgfra positive cells. It simply states in the manuscript that they are positive. However, we see in

the representative image that the FAPs are not 100% positive, nor could the MuSCs be 100% positive.”

We amended the text to say that “freshly isolated” MuSCs and FAPs were stained. In addition, we have included the results of the quantifications of the PAX7 and PDGFRA stains in the text.

- 4) *“The FAPs TA cross section staining with PDGFRA is unconvincing. There is far too much non-specific staining to make any conclusion. Another antibody must be used.”*

This staining is challenging to perform due to the need for a protocol that works across three species. We have replaced the image of the PDGFRA stain with a different image showing less non-specific staining. Unfortunately, we were unable to identify a second cross-reacting antibody against mouse lemur PDGFRA.

- 5) *“In the DENSPM treatment it should be made clear in the results section that the mice were given multiple doses of the compound after transplantation. Otherwise, it makes it seem as if the cells were only treated prior to transplantation. Also, the concentration used should be added to the figure legend or the figure itself and not just in the methods section (Figure 2L)”*

We have amended the text to highlight the extra dose of DENSPM. We have amended the figure legend to include the concentration of DENSPM (2.5 mM of DENSPM (Tocris; #0468)).

- 6) *“Representative image for EDU staining of spermidine-treated cells (Figure 2M) and the concentration of spermidine added should be in the figure or the legend”*

We have included representative images for the EdU staining of Spermidine-treated MuSCs (Figure 2M). We have amended the figure legend to include the concentration of Spermidine used (100 nM of Spermidine (Sigma-Aldrich; # S0266)).

REVIEWER #3:

“I have reviewed the revised manuscript and the authors have satisfactorily addressed all the previous comments. The changes made improve the clarity and quality of the manuscript. I believe the paper is now suitable for publication in Nature Communications. I have no further comments and recommend accepting the manuscript for publication.”

We thank the Reviewer for recommending the manuscript for publication.